# Reward expectation enhances action-related activity of nigral dopaminergic and two striatal output pathways

Alain Rios [1,8✉], Satoshi Nonomura[1,2,8], Shigeki Kato[3], Junichi Yoshida [4], Natsuki Matsushita [5], Atsushi Nambu [6], Masahiko Takada [2], Riichiro Hira[1], Kazuto Kobayashi [3], Yutaka Sakai[7], Minoru Kimura[7] & Yoshikazu Isomura [1,7✉]

Neurons comprising nigrostriatal system play important roles in action selection. However, it remains unclear how this system integrates recent outcome information with current action (movement) and outcome (reward or no reward) information to achieve appropriate subsequent action. We examined how neuronal activity of substantia nigra pars compacta (SNc) and dorsal striatum reflects the level of reward expectation from recent outcomes in rats performing a reward-based choice task. Movement-related activity of direct and indirect pathway striatal projection neurons (dSPNs and iSPNs, respectively) were enhanced by reward expectation, similarly to the SNc dopaminergic neurons, in both medial and lateral nigrostriatal projections. Given the classical basal ganglia model wherein dopamine stimulates dSPNs and suppresses iSPNs through distinct dopamine receptors, dopamine might not be the primary driver of iSPN activity increasing following higher reward expectation. In contrast, outcome-related activity was affected by reward expectation in line with the classical model and reinforcement learning theory, suggesting purposive effects of reward expectation.

[1] Department of Physiology and Cell Biology, Graduate School of Medical and Dental Sciences, Tokyo Medical and Dental University (TMDU), Tokyo 113-8510, Japan. [2] Center for the Evolutionary Origins of Human Behavior, Kyoto University, Aichi 484-8506, Japan. [3] Department of Molecular Genetics, Institute of Biomedical Science, Fukushima Medical University, Fukushima 960-1295, Japan. [4] Dominick P. Purpura Department of Neuroscience, Albert Einstein College of Medicine, Bronx, NY 10461, USA. [5] Division of Laboratory Animal Research, Aichi Medical University, Aichi 480-1195, Japan. [6] Division of System Neurophysiology, National Institute of Physiological Sciences and Department of Physiological Sciences, SOKENDAI, Aichi 444-8585, Japan. [7] Brain Science Institute, Tamagawa University, Tokyo 194-8610, Japan. [8] These authors contributed equally: Alain Rios, Satoshi Nonomura. ✉email: aardphy2@tmd.ac.jp; isomura.phy2@tmd.ac.jp

Dopamine neurons (DANs) in the substantia nigra pars compacta (SNc) play an important role in movement and reward signaling[1,2]. Recent studies report that the phasic activity of SNc DANs changes synchronously with action onset, and that optogenetic manipulation of this activity instantaneously affects the initiation of action[2,3], thereby suggesting that phasic DAN activity is involved in immediate action modulation. It is currently believed that SNc DANs influence the dorsal striatum to perform this function[4]. The majority of striatal neurons consist of projection neurons (SPNs)[5], which give rise to the two intrinsic pathways of the basal ganglia, the direct and indirect pathways (dSPNs and iSPNs, respectively)[1,6,7]. In general, these two striatal pathways are hypothesized to exert an antagonistic effect on motor control through the excitation and inhibition of the downstream thalamus and cortex[1,6,8], in agreement with reports on their antagonistic involvement in locomotor activity[9], reinforcement[10,11], and drug sensitization[12]. Although it is well acknowledged that dopamine depolarizes and hyperpolarizes dSPNs and iSPNs through dopamine D1 and D2 receptors, respectively[13], dSPNs and iSPNs are now known to be activated concurrently to produce coherent sequences of voluntary actions and to convey similar reward information[14–17]. In addition, we recently reported that in goal-directed action selection, outcome signals are differentially represented via dSPNs (encoding reward outcome) and iSPNs (encoding no-reward outcome) in the rat dorsomedial striatum (DMS), although showing concurrent activation during movement initiation[18]. However, there is still a debate on how the phasic activity of SNc dopamine and dorsal striatum neurons integrates the information on recently obtained rewards with the current action and outcome information to achieve the subsequent action appropriately (i.e. action selection guided by reward acquisition in past trials).

On the other hand, many lines of evidence showed that phasic DAN activity is related to outcome information processing rather than action execution, particularly encoding reward prediction error (RPE) to learn the most valuable action through trial and error[19,20]. The models explaining the roles of dopamine in outcome information processing, in addition to action execution, propose that different functions depend on the timing of the DAN activity change (e.g. the phasic activity of DANs is mainly related to the RPE signal, while persistent activity changes are related to action execution)[21,22]. Besides the DANs, ~30% of the neurons in the SNc are GABAergic and may be critically involved in the regulation of DAN activity[23]. Moreover, the nigro-striatal system exhibits a medio-lateral functional and anatomical segregation[24]. In rodents, the medial and lateral regions of the SNc (hereafter, mSNc and lSNc, respectively) send distinct reward-related dopamine signals to the DMS or dorsolateral striatum (DLS)[25–27]. In monkeys, the mSNc encodes motivational value, whereas the lSNc encodes motivational salience[28]. It is also known that the target DMS and DLS neurons have different functional properties[29,30], receiving topographical inputs from SNc and various areas of the cerebral cortex[31,32]. In particular, DMS is associated with goal-directed behavior[33,34], while DLS is associated with the formation of habitual behavior[34,35].

Taken together, a pivotal question arises as to how the different neuron populations of the nigro-striatal system participate in the integration of previous reward information with current action- and outcome-related activity. To address this question, we trained rats under head-fixed conditions to complete a reward-based choice task[18]. The rats had to select and execute the adequate action based on the presence or absence of a reward in past trials. We examined the action- and outcome-related activity of optogenetically identified striatal and nigral projection neurons during task performance, together with measurement of local dopamine release in the dorsal striatum.

## Results

**Contribution of recent outcomes to current action choice strategy**. To study how the action- and outcome-related neuronal activity and nigrostriatal dopamine transmission relate to recently obtained rewards during decision making, we trained adult wild type (WT) rats ($n = 6$) and transgenic rats expressing Cre recombinase under Tac1 ($n = 7$), D2R ($n = 6$), or TH ($n = 7$) promotor to perform a push/pull choice task based on probabilistic reward[18] (Fig. 1a). After holding a lever in the center position for more than 300 ms, a visual cue (Go-cue) was provided, serving as a signal for the rats to either pushing or pulling a lever based on the reward probability assigned to each action. 70% of correct (preferable) responses (e.g., push in the case of Fig. 1a) and 10% of incorrect (unpreferable) responses (pull in the case) were followed by a high-tone sound (10 kHz, 60 dB, 0.3 s) signaling a reward outcome. The reward (water) was then delivered with a delay of 0.3 s. Conversely, the remaining responses were followed by a low-tone sound (4 kHz, 60 dB, 0.3 s) indicating a no-reward outcome. In this case, no water was delivered. The high-probability rewarded choice (e.g., push in Fig. 1a) was changed after at least 30 correct choices and a 79% correct rate in the last ten trials. Individual rats mastered the task (error rate <15%) within 25 days of training. On average, the rats adapted to select actions associated with high reward probability within 15 trials and then continued to select them in more than 79% of subsequent trials (Fig. 1b, c). Next, we evaluated the effects of recent outcomes on the upcoming action choice. The contribution of recent outcomes to the action choice strategy can be estimated by logistic regression[36]. We used a regression model in which the probability of staying or switching selection in the next trial was determined by the rats' recent outcomes (Fig. 1d). The contribution of recent outcomes declined with the passage of trials. Rewards in the previous five trials had a significant effect on choices in the upcoming trial, persuading the animal to stay with the same choice (positive regression coefficients; $\beta = 0.27 \pm 0.36$, $p = 0.0015$ with the fifth-past trial). The absence of reward in the previous two trials promoted switching (negative regression coefficients; $\beta = -0.18 \pm 0.26$, $p = 0.04$ with the second-past trial). Given that the choice on a given trial was influenced by the outcomes of the preceding five trials, we quantified the average reward experience as a proportion, referred to as the 'reward rate' (Fig. 1b, green). Specifically, the reward rate is calculated as the mean number of rewards received in the last five trials (without including current trial), i.e., the number of rewards divided by five. This yields a value between 0 and 1, with 0 indicating no rewards received in the last five trials, and 1 indicating that all five last trials were rewarded. By using this proportional scale, we can accurately represent the recent reward experience for each trial. We observed that the rats implemented a win-stay lose-switch strategy in which rewards served as evidence to stay with the same choice, and no-reward outcomes promoted switching (Fig. 1e). The reaction time (time from Go-cue to movement onset; $218 \pm 61$ ms overall) and lever movement time ($143 \pm 224$ ms) were also significantly correlated with reward rate (Fig. 1f). These results suggest that the rats used the outcome information from up to the recent five trials to guide their behavior. Based on the contribution of the reward rate to the next choice, from here on, we will evaluate how the reward rate correlates with dopamine release and activity of neurons in the nigral and striatal regions.

**Striatal dopamine release correlated with reward rate**. To explore how dopamine release in dorsal striatum related to the reward rate, we injected AAV vector into DMS or DLS to express dLight 1.3b and monitored the fluorescence signal around different

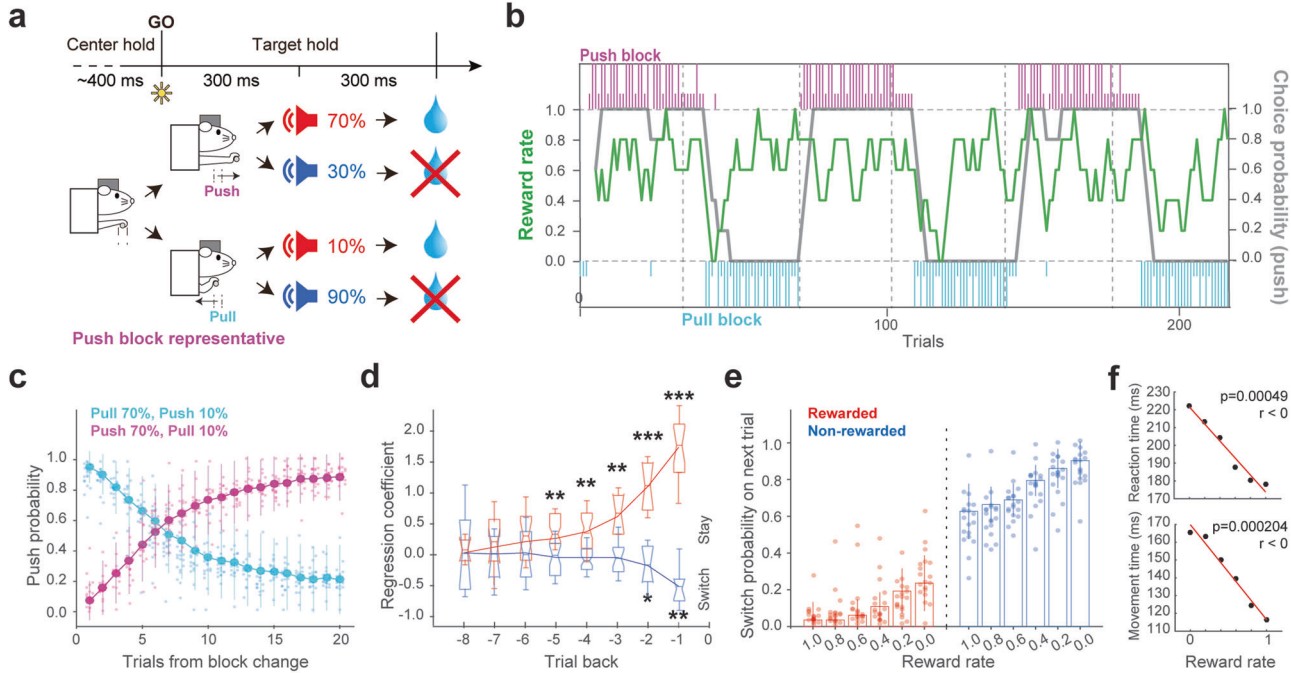

**Fig. 1 Impact of recently obtained rewards on action choice strategy. a** Schematic of a push/pull choice task performed by head-fixed rats based on probabilistic reward. Rats chose to either push or pull the lever depending on the block condition. Reward delivery or absence was instructed by different tones. The timeline represents a push block's trial, indicating the center holding, lever movement, and outcome timing. The probability of high-tone sound indicating reward delivery was 70% after a correct choice and 10% after an incorrect choice. The tone probability was reversed after at least 30 correct trials and a 79% correct rate in the last 10 trials. See "Methods" section for details. **b** Representation of individual choices across blocks. Magenta and cyan vertical lines indicate individual choices in push or pull blocks, respectively (long, rewarded; short, non-rewarded). The gray line indicates the probability of a push choice (running average of the last five choices). The green line indicates the average rewarded choices in the past five trials (reward rate). **c** Number of trials needed to correctly change the choice after a block change. Magenta and cyan lines indicate the probability of a push choice after changing to a block with higher push or pull reward probability, respectively (20 rats; 58 sessions). Error bars represent standard deviation. Each dot represents one rat average. **d** Choice in upcoming trials was determined by the outcomes of previous trials. Contributions of rewarded outcomes (red) in the previous five trials and non-rewarded (blue) outcomes in the previous trials on choices in the current trial, as derived from logistic regression (58 sessions, 659 trials/session on average); *$p < 0.05$. Error bars represent standard deviation. On each box, the central mark indicates the median, the bottom and top edges of the box indicate the 25th and 75th percentiles, respectively. The whiskers show the extreme data points not considered outliers. Notches show the 95% confidence interval. **e** Effect of the reward rate on the next choice. The probability of switching the choice in the next trial after either rewarded or non-rewarded outcome selection in the current trial, with statistical dependence on the reward rate (average ratio of rewards obtained in the last 5 trials). Error bars represent standard deviation. Each dot represents one rat average. **f** Reaction time and lever movement time correlated with reward rate. Error bars represent standard error of the mean (SEM).

task-related events (Go-cue, movement onset, and outcome tone: The movement onset occurred after a variable delay following the Go-cue, i.e., reaction time) using fiber photometry (Fig. 2a and Supplementary Fig. 1). The dLight signal at 470 nm excitation showed robust task-related changes (Fig. 2b). In contrast, no change was observed in the control 405 nm excitation (Fig. 2c). The population data showed a rapid robust signal increase after the Go-cue, with an ascending phase coincident with movement onset, illustrating the delay between the Go-cue and the actual movement initiation (Fig. 2d, e). This action-related signal activation was similar in the DMS and DLS (Fig. 2g, h), independent of the current trial outcome. We also observed a characteristic signal change in the DMS and DLS in response to outcome: a clear signal increase after the reward tone and a signal decrease after the no-reward tone ('dopamine dip'). However, there was a difference in the time course of their changes: in the DLS dopamine signal exhibited a sharp response and a return to baseline within 500 ms (Fig. 2i), whereas in the DMS signal showed a slower and long-lasting change (Fig. 2f). Because we observed different movement-related dopamine signal changes during push and pull selections (Supplementary Fig. 1), we used only the sessions that exhibited a preferred activity modulation during either push or pull selections. To evaluate how the dopamine signal correlated with recently

obtained rewards, we used the average number of rewards over the last five trials (reward rate), based on the behavioral analysis (Fig. 1). We evaluated the reward rate correlation with the population dopamine signal at different time windows: immediately after the Go-cue, after the lever movement onset, and after the outcome tone, differentiating between rewarded and non-rewarded trials (Fig. 2d–i, insets; Supplementary Fig. 2; see also Supplementary Table 1). After the movement onset, the DMS and DLS dopamine signals were positively correlated with reward rate (Fig. 2e, h and Supplementary Fig. 2a, b). However, during the outcome period, the signal was stronger when the reward rate was lower, being consistent with positive RPE (Fig. 2f, i and Supplementary Fig. 2a, b).

**Identified SNc DAN activity correlated with reward rate.** It was reported that some patterns of local dopamine release are inconsistent with the firing activity of dopamine cells[37], suggesting that the examination of firing patterns of dopamine cells was needed for understanding the role of dopamine signals in reward information processing. We recorded the neuronal activity in the SNc of task-performing TH-Cre rats under a head-fixed condition. We isolated a total of 1523 neurons during task performance. These neurons were further classified into putative

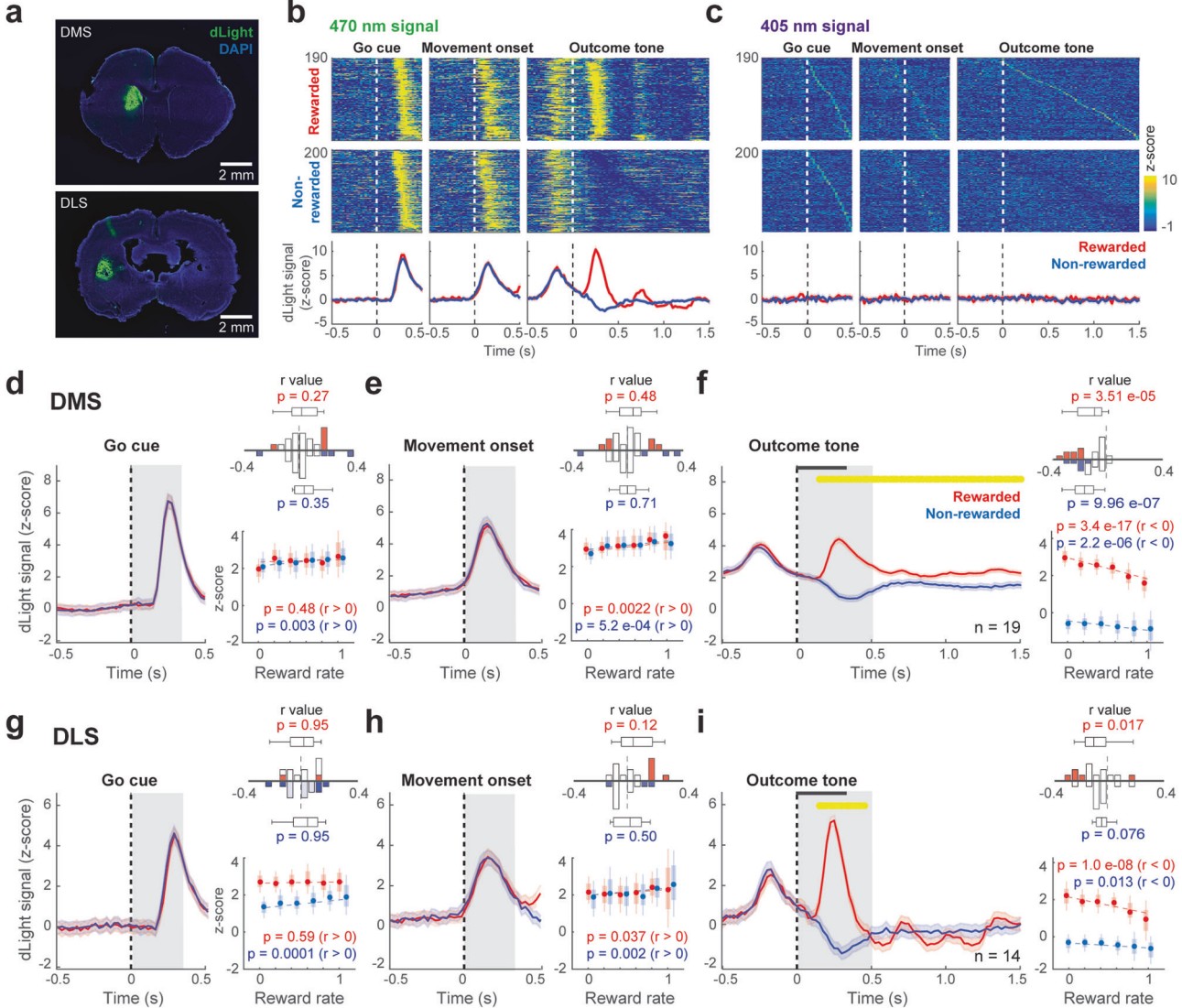

**Fig. 2 Correlation of striatal dopamine release with reward rate. a** Local expression of the genetically encoded dopamine sensor dLight1.3b in DMS and DLS. **b** Representative event-aligned dLight fluorescence signals in rewarded and non-rewarded trials. The biphasic dLight fluorescence changes occurred in the Go cue, movement, and outcome periods at 470 nm excitation. Note the same first peak at 470 nm was seen during Go cue and movement and before outcome tone. The signal was differentiated into another peak (red) and dip (blue) only after the outcome appeared. **c** Representative event-aligned isosbestic signal at 405 nm in rewarded and non-rewarded trials. **d** dLight signal in the Go cue period in DMS. The yellow horizontal line above traces represents a significant difference ($p < 0.01$, bin 20 ms; $t$-test) between rewarded (red) and non-rewarded (blue) responses. Shading area represents the standard error. Horizontal black bars indicate outcome tone duration. Line plot shows the statistical dependence of population dLight signal on reward rate in current rewarded (red) and non-rewarded (blue) trials. All error bars represent SEM. On each box, the central dot indicates the median, the bottom and top edges of the box indicate the 25th and 75th percentiles, respectively. The whiskers show the extreme data points not considered outliers. Histograms in insets show the distribution of the correlation coefficient ($r$) of dLight signal correlation with reward rate for individual sessions. Colored bars represent significant correlation ($p < 0.05$). The central mark on the box plot indicates the median, the bottom and top edges of the box indicate the 25th and 75th percentiles, respectively. The whiskers show the extreme data points not considered outliers; p value indicates the comparison between the distribution of $r$ values and 0 (Wilcoxon rank-sum test). **e** Same as **d** for movement onset window. **f** Same as **d** for outcome window. **g** dLight signal in the Go cue period in DLS. **h** Same as **g** for movement onset window. **i** Same as **g** for outcome window.

DANs ($n = 502$) and GABAergic interneurons ($n = 1021$) according to a clear bimodal distribution of their spike duration (we cannot discard the possibility that the narrow spikes may correspond to neighboring SNr neurons, Supplementary Fig. 3a). We then further selected neurons that showed a significant movement-related and/or outcome-related response (see "Methods" section, DANs: $n = 212$; INs: $n = 138$; see Supplementary Fig. 4 for representatives). The ongoing spike rates of the putative GABAergic neurons ($3.01 \pm 6.2$ Hz) were significantly higher than those of putative DANs ($1.14 \pm 2.25$ Hz; $p = 1.42e^{-04}$, Wilcoxon

rank-sum test), consistent with previous reports[37]. To confidently differentiate between the SNc DANs projecting to the DMS and those projecting to the DLS (SNc-DMS and SNc-DLS DANs, respectively), we used an optogenetically evoked spike collision test with antidromic stimulation of either DMS or DLS while recording SNc neurons of TH-Cre rats[38]. We injected AAV2-EF1α-Flex-ChRWR/Venus into the left SNc of TH-Cre rats. The expression of Venus in the striatum and the coexpression of Venus and TH in the SNc DANs were confirmed histologically (Fig. 3a). Figure 3b shows representative tetrode traces of

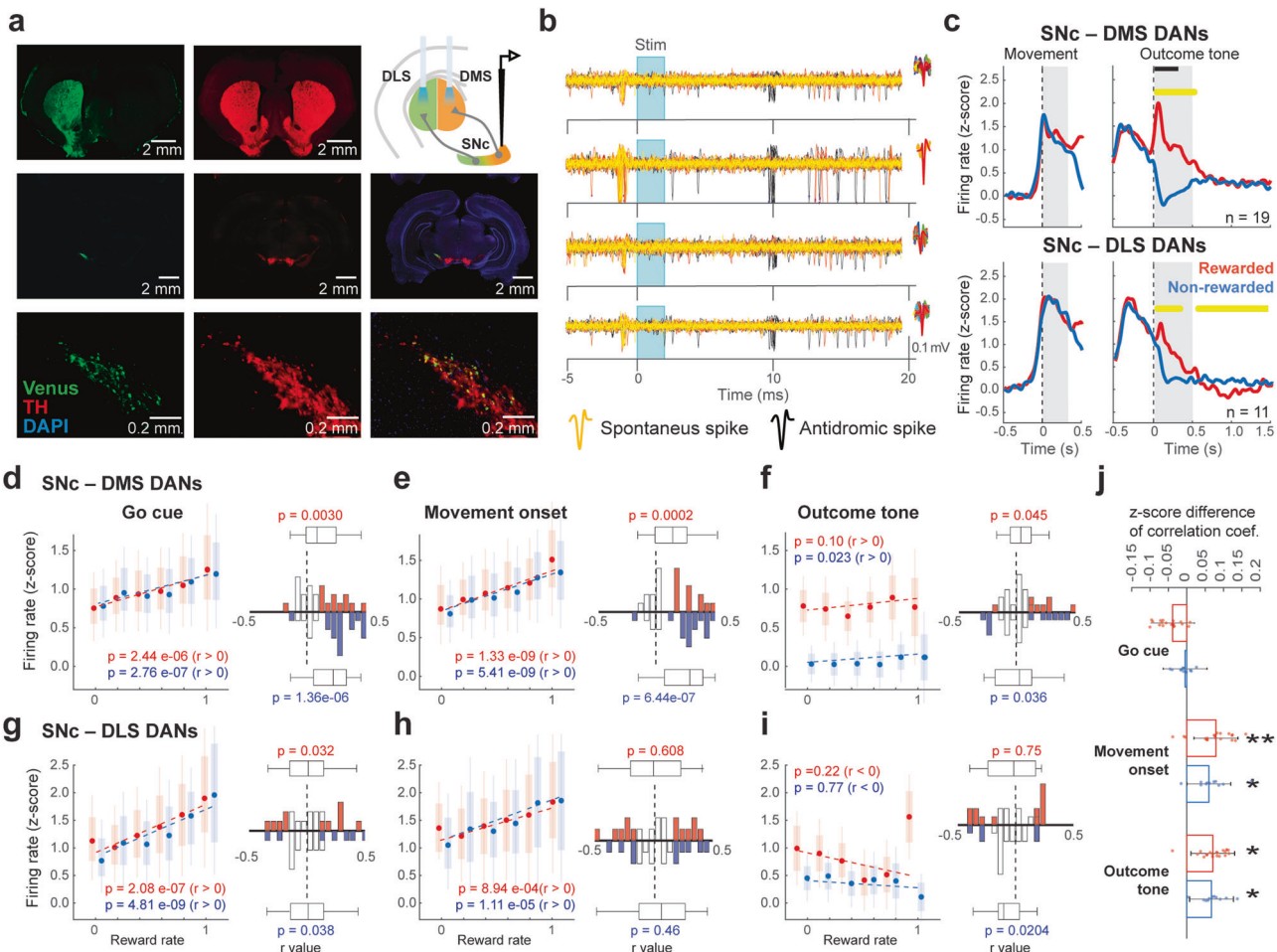

**Fig. 3 Correlation of identified SNc DAN activity with reward rate. a** Selective expression of ChRWR/Venus in SNc DANs. Schematic diagram of the vector injection site (AAV2-EF1α-Flex-ChRWR/Venus) in the left SNc, and the stimulation site via the optical fiber in DMS and DLS for optogenetic spike collision test (top-right). Left panels show Venus expression in the left striatum (top), left midbrain (middle), and SNc neurons (bottom). Center panels show tyrosine hydroxylase (TH) staining in the striatum (top), midbrain (middle), and SNc neurons (bottom). Right middle and bottom panels show overlapping of Venus and TH expression in the SNc neurons. **b** Identification of striatum projecting DANs by spike collision test. Representative spike activity in one tetrode around optical stimulation. Black traces represent antidromic spikes evoked by optical stimulation (cyan). Yellow traces show the absence of the antidromic spike after optical stimulation triggered by spontaneous spikes, confirming a successful spike collision. The spike waveforms are shown in red. **c** Response of DMS- and DLS-projecting DANs aligned with movement onset and outcome (rewarded or non-rewarded) tones. The yellow horizontal line represents a significant difference ($p < 0.01$, bin = 20 ms; $t$-test) between rewarded (red) and non-rewarded (blue) responses. Horizontal black bars indicate outcome tone duration. **d** Reward rate correlation of SNc-DMS with Go cue period activity during rewarded (red) and non-rewarded (blue) trials. All error bars represent SEM. On each box, the central dot indicates the median, the bottom and top edges of the box indicate the 25th and 75th percentiles, respectively. The whiskers show the extreme data points not considered outliers. The histograms show $r$ value distribution of individual neurons; colored bars represent significant correlation. The central mark on the box plot indicates the median, the bottom and top edges of the box indicate the 25th and 75th percentiles, respectively. The whiskers show the extreme data points not considered outliers; $p$ value indicates the comparison between the distribution of $r$ values and 0 (Wilcoxon rank-sum test). **e** Same as **d** for movement onset period. **f** Same as **d** for outcome period. **g** Reward rate correlation of SNc-DLS during Go cue period. **h** Same as **g** for movement onset period. **i** Same as **g** for outcome period. **j** Comparison of the strength of the neural activity correlation with reward rate between SNc-DMS and SNc-DLS. Difference of $z$-score of the correlation coefficients using Fisher's $z$-transformation. Each dot represents one neuron. The error bars correspond to the confidence interval; *$p < 0.05$, **$p < 0.01$; $\alpha = 0.05$. Fisher's $z$-test.

antidromically evoked spikes (black) and their disappearance due to collision with spontaneous spikes (yellow) in an SNc recorded neuron. We identified 33 SNc-DMS DANs and 23 SNc-DLS DANs. The latency of the antidromically evoked spikes in SNc-DMS DANs and SNc-DLS DANs was $14.5 \pm 4.6$ ms and $12.6 \pm 3.8$ ms, respectively (Supplementary Fig. 5a, b). All DANs exhibited a wide spike waveform (SNc-DMS: $1.05 \pm 0.16$ ms; SNc-DLS: $0.96 \pm 0.12$ ms) and a low spike rate (SNc-DMS: $2.02 \pm 1.60$ Hz; SNc-DLS: $1.92 \pm 1.45$ Hz; Supplementary Fig. 5c, d). We then further selected neurons that showed a significant movement-related and/or outcome-related response (see Methods), 23 SNc-DMS and 22 SNc-DLS DANs were obtained in both.

To evaluate how the neuronal activity variated depending on the reward rate, we used only the neurons that exhibited preferred activity modulation during either push or pull selection. Nineteen SNc-DMS and 11 SNc-DLS DANs exhibited higher activation during either a push or pull selection. These identified task-related neurons showed a mediolateral segregation (Supplementary Fig. 5e, f). Some neurons classified as GABAergic interneurons exhibited some response to the light stimulation. However, this response did not comply with the criteria used for the analyzed neurons (i.e., fixed latency, small jitter, collision test). So, we did not include these data of GABAergic interneurons in the analysis.

Both SNc-DMS and SNc-DLS DANs exhibited robust phasic activation after the Go-cue. Consistent with the activity of the putative SNc DANs (Supplementary Fig. 3b), the identified striatum-projecting DANs showed a characteristic pattern in response to reward and no-reward outcomes (Fig. 3c), i.e., increased activity after the reward tone and suppression after the no-reward tone. The latency of the DANs activity was shorter than that of the dopamine signal response (Fig. 3c vs. Fig. 2e, h), as expected considering the time needed for dopamine release and uptake[39]. Contrastingly, putative GABAergic interneurons displayed a phasic short-latency activation followed by prolonged activation after the reward tone, and a slower activation followed by prolonged suppression in response to no-reward tone (Supplementary Fig. 3b). SNc-DMS DANs showed higher activation than SNc-DLS DANs in response to a reward ($p = 2.6e^{-21}$, rank-sum test). Additionally, the activity suppression after no-reward in SNc-DLS DANs was smaller than in the SNc-DMS DANs ($p = 0.005$, rank-sum test). This response pattern agrees with previous reports[40], where DLS-projecting neurons presented activation in response to a reward but showed a lack of a dopamine dip after reward omission.

Next, we evaluated how the neuronal activity correlated with the reward rate (Supplementary Fig. 3c). Consistent with the putative DANs (Supplementary Fig. 3d), the SNc-DMS and SNc-DLS DAN activity after Go-cue and movement onset was positively correlated with reward rate (Fig. 3d, e, g, h). Additionally, only the SNc-DMS DANs exhibited a positive correlation with reward rate during the outcome period (Fig. 3f), whereas the SNc-DLS DANs had a tendency for a negative correlation (Fig. 3i). The correlation of the movement-related SNc-DMS DANs was significantly stronger than that of the SNc-DLS DANs (rewarded: $p = 0.009$; non-rewarded: $p = 0.047$, Fisher's z-test, Fig. 3j). Meanwhile, no significant difference in the reward rate correlation of movement-related activity was found between rewarded and no rewarded trials in SNc-DMS or SNc-DLS DANs (Fig. 3j). These results suggest that the different neuronal populations in the SNc may have a similar movement-related activity correlation with reward rate, consistent with change in state value[37,41]. On the other hand, the reward rate signaling during the outcome period differed across population types and their mediolateral distribution (Fig. 3f, i). Additionally, the correlation of the outcome-related activity of SNc-DMS DAN with the reward rate notably differed from the dopamine signal in the DMS (Fig. 3f vs. Fig. 2f, inset), where the SNc-DMS DAN exhibited a higher activity as the reward rate was higher, while the dopamine signal exhibited the opposite pattern. These results indicate that dopamine release may convey different information to dopamine neuron firing, supporting the idea that dopamine local release dynamics may be controlled in different ways[37].

**Identified dSPN and iSPN activities correlated with reward rate.** Previous studies have shown a high degree of specificity in the interactions between different cortical and thalamic pathways according to postsynaptic striatal cell types[42]. Our results showed that the action-related DAN activity and local dopamine release increased with higher reward expectation. The classical model for direct (dSPN) and indirect pathway neurons (iSPN) in the striatum postulates excitatory and inhibitory effects of dopamine on dSPN and iSPN, respectively[6]. Therefore, a higher dopamine concentration should exert opposite effect on dSPNs and iSPNs acting through dopamine D1 and D2 receptors, respectively. We performed DMS and DLS recordings and selectively tracked the activity of dSPNs (expressing D1 receptors) and iSPNs (expressing D2 receptors) using AAV2-EF1α-Flex-ChRWR/Venus or AAV2-Syn-Flex-rcChrimsonR-tdTomato injections in Tac1-Cre

(for dSPN) or Drd2-Cre (for iSPN) transgenic rats. We confirmed that dSPNs and iSPNs project to the SNr (striatonigral neurons) and GPe (striatopallidal neurons), respectively (Figs. 4a and 5a).

We isolated a total of 2002 neurons from the dorsal striatum and classified them using three parameters: ongoing spike rate, spike duration and coefficient variation (CV) of interspike interval (ISI) (Supplementary Fig. 6a). Three clearly separate clusters were formed, assignable to the previously described striatal neuron subpopulations, i.e., SPNs ($n = 716$), tonically active neurons (TANs, $n = 1224$), and fast-spiking interneurons (FSIs, $n = 62$; Supplementary Fig. 6b): we obtained the task-related neurons using the same criteria as for the SNc DANs (SPNs, $n = 138$; TANs, $n = 274$; and FSIs, $n = 17$). Putative DMS-SPNs and DLS-SPNs exhibited a phasic activation in response to the Go-cue and coincided with the movement onset (Supplementary Fig. 6c). Movement- and outcome-related activities of SPNs were positively correlated with the reward rate (Supplementary Fig. 6d). TANs in both DMS and DLS exhibited a subtle activation after the Go-cue and movement-onset, which was positively correlated with the reward rate in non-rewarded trials. FSIs showed activation after Go-cues and movement onset, which was also positively correlated with reward rate in non-rewarded trials. These results show that the positive reward rate correlation with the action-related activity was predominant in SPNs and interneurons.

For the identification of the dSPNs, we confirmed the responsiveness to single light pulses delivered next to the recording sites in the DMS or DLS, selecting only those neurons that exhibited short latency (<8 ms), small jitter (<1 ms), and a high waveform correlation between light-evoked and spontaneous spikes (Fig. 4b, c; see "Methods" section). We identified 50 task-related dSPNs in the DMS and 28 dSPNs in the DLS. We used only those neurons that exhibited a preferred activation during the movement period (DMS-dSPNs: $n = 44$; DLS-dSPNs: $n = 22$). dSPNs showed robust activation during the movement onset in both the DMS and DLS (Fig. 4d). DMS- and DLS-dSPNs showed a higher activation in response to rewarded than non-rewarded outcomes (Fig. 4d), in agreement with previous results from our laboratory[18]. The movement-related activity of DMS-dSPNs showed positive correlation with reward rate in both rewarded and no-rewarded trials (Fig. 4f). That of DLS-dSPNs exhibited positive correlation with reward rate only in no-rewarded trials (Fig. 4i and Supplementary Fig. 2e, f). The outcome-related activity of DLS-dSPNs after no-reward was positively correlated to reward-rate (Fig. 4j).

For the identification of the iSPNs, we evaluated the responsiveness of the striatopallidal neurons to single light pulses delivered from the target GPe (Fig. 5b, c). We matched the light-evoked spikes to the spontaneous spikes, and thereby distinguished iSPNs from D2-positive TANs, identifying 31 task-related iSPNs from DMS and 32 iSPNs from DLS. We analyzed only those neurons that exhibited a preferred activation during the movement period in accordance with dopamine analysis (DMS-iSPNs, $n = 22$; DLS-iSPNs, $n = 20$). iSPNs showed robust activation during the movement period in both DMS and DLS (Fig. 5d). Overall, peak latencies of movement-related firing activity were similar among different groups of nigral and striatal neurons except for DMS–SNc neurons (Supplementary Fig. 7). Importantly, the activation of those neurons preceded dopamine release in the striatum (e.g., Fig. 2e. h vs. Figs. 3c, 4d, and 5d), suggesting possible common inputs other than dopamine.

In contrast to dSPNs response, DMS-iSPNs exhibited activation after no-reward outcomes and suppression after reward outcomes (Fig. 5d and Supplementary Fig. 4d) consistent with our previous observation[18]. However, DLS-iSPN activity did not differ between rewarded and non-rewarded trials at a population

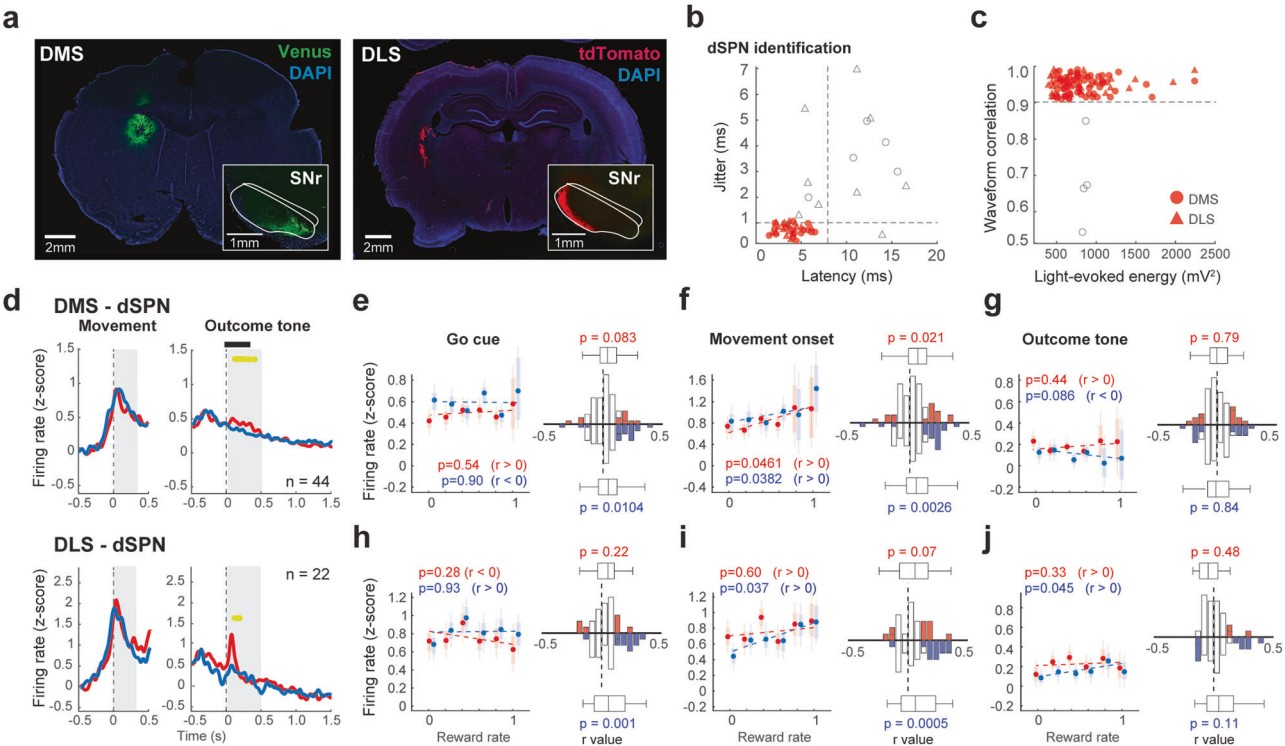

**Fig. 4 Correlation of identified dSPN activity with reward rate. a** Local expression of ChRWR/Venus or rcChrimsonR-tdTomato in DMS or DLS of Tac1-Cre. Insets show the striato-nigral projections into the SNr. **b** Light-evoked spikes elicited by the light pulse. Latency and jittering criteria for successful identification (red symbols). **c** Correlation coefficient for four-channel patterns of mean spike waveforms between spontaneous and light-evoked spikes plotted against the energy of light-evoked responses for each neuron. **d** Responses of optogenetically identified dSPNs of DMS and DLS aligned with movement onset and outcome (reward or no-reward) tones. The yellow horizontal line represents a significant difference ($p < 0.01$, bin = 20 ms; $t$-test) between rewarded (red) and non-rewarded (blue) responses. Horizontal black bars indicate outcome tone duration. **e** Reward rate correlation of DMS-dSPNs with Go cue period activity during rewarded (red) and non-rewarded (blue) trials. All error bars represent SEM. On each box, the central dot indicates the median, the bottom and top edges of the box indicate the 25th and 75th percentiles, respectively. The whiskers show the extreme data points not considered outliers. The histograms show $r$ value distribution of individual neurons; colored bars represent significant correlation. The central mark on the box plot indicates the median, the bottom and top edges of the box indicate the 25th and 75th percentiles, respectively. The whiskers show the extreme data points not considered outliers; p value indicates the comparison between the distribution of $r$ values and 0 (Wilcoxon rank-sum test). **f** Same as **e** for movement onset period. **g** Same as **e** for outcome period. **h** Reward rate correlation of DLS-dSPNs during Go cue period. **i** Same as **h** for movement onset period. **j** Same as **h** for outcome period.

level, although some neurons showed higher activation after no reward tone (Supplementary Fig. 4f). The activity of both DMS- and DLS- iSPNs populations was positively correlated with reward rate during movement and outcome periods (Fig. 5e–j and Supplementary Fig. 2g, h). The DMS-iSPNs exhibited the strongest correlation with reward rate in both movement- and outcome related activity among DMS and DLS SPNs (Fig. 5k).

## Discussion
In the present study, we investigated how previous rewards information is integrated into action- and outcome-related activity through distinct types of projection neurons and interneurons in SNc and dorsal striatum according to their medio-lateral topography (Fig. 6). We used rats under a head-fixed condition performing a reward-context-dependent lever push/pull choice task (Fig. 1), and examined local dopamine release, DMS- and DLS-projecting SNc DAN activity, and DMS and DLS SPN and interneuron activity. To identify different neuron types, we applied optogenetics and electrophysiological techniques, accomplishing high spatial and temporal resolution at a cellular level[18,43–45].

In our study, the action-related (both Go-cue- and movement-related) local dopamine release (Fig. 2) and activity of both DMS-

and DLS-projecting SNc DANs (Fig. 3) were enhanced by reward expectation reflected by reward rate (Fig. 6; green, small upward arrows indicate enhancement). Additionally, the action-related activity of most types of DMS and DLS neurons also exhibited similar positive correlation with the reward rate (Figs. 4–6). According to the classical model[1,6,10], dopamine in the striatum exerts an excitatory and inhibitory effect on dSPNs and iSPNs, respectively, thereby regulating the activity balance of the downstream thalamus and cortex antagonistically. However, it was reported that the concomitant action-related activity of dSPNs and iSPNs may convey similar reward information[14,18], which cannot be explained by the classical model alone. In this study, the action-related activity of unidentified SPN populations was positively biased toward higher reward rate in both the DMS and DLS (Fig. 6), suggesting an enhancing effect of reward expectation on their activity. In particular, the action-related activities of the identified DMS- and DLS-iSPNs were positively correlated with the reward rate, suggesting a potential dissociation between iSPN activity and dopamine release timing or that reward expectation can override such dopamine effects all through mediolateral nigrostriatal and striato-pallidal systems (Fig. 5). The robustness observed in the iSPN activity could potentially offset the inhibitory influence of dopamine on these neurons, such as the anticipated inhibitory effect mediated

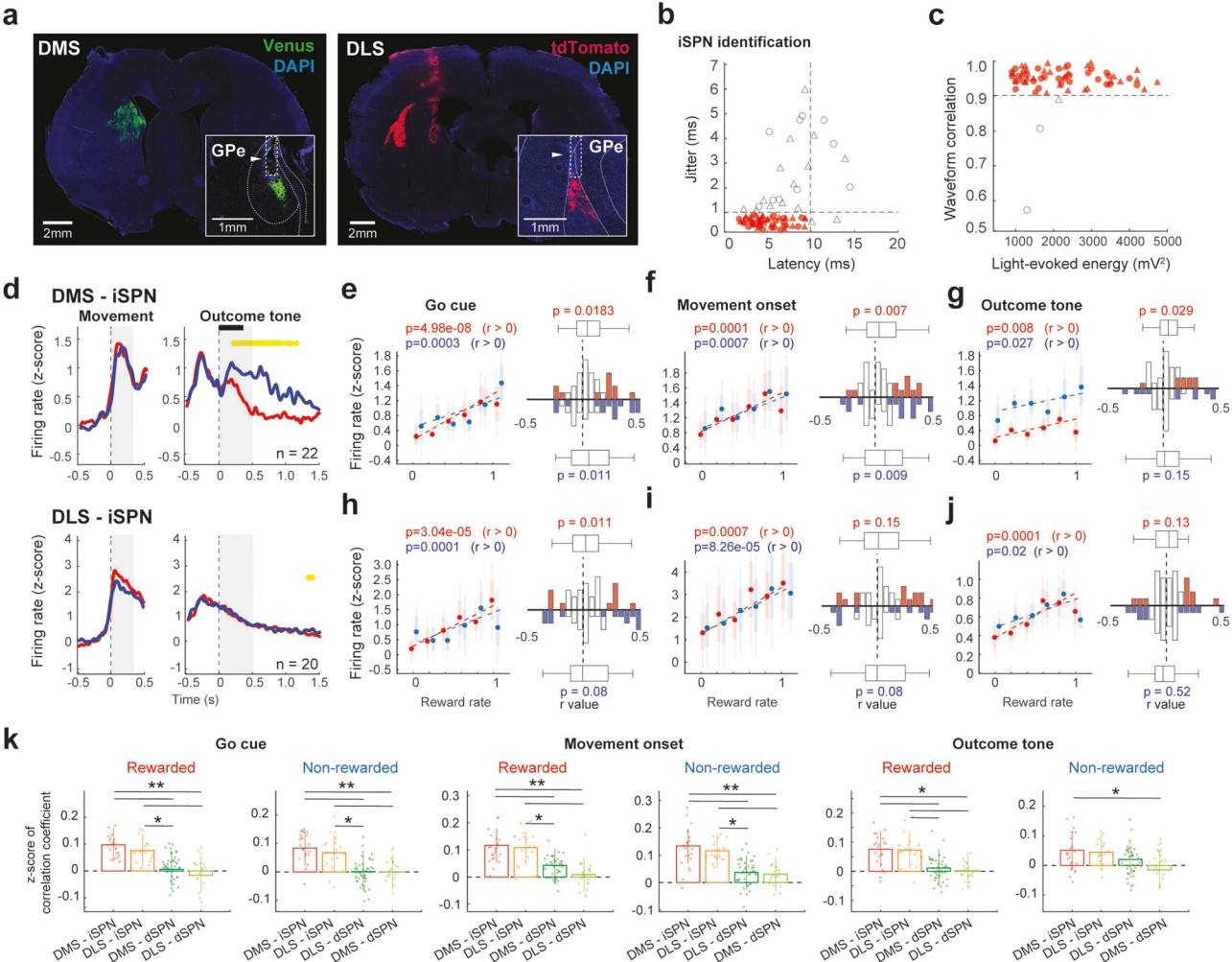

**Fig. 5 Correlation of identified iSPN activity with reward rate. a** Local expression of ChRWR/Venus or rcChrimsonR-tdTomato in DMS or DLS of Drd2-Cre rats. Insets show the striato-pallidal axons projecting into the GPe. Arrowheads show the tip of the optical fiber. **b** Light-evoked spikes elicited by the light pulse. Latency and jittering criteria for successful identification (red symbols). **c** Correlation coefficient for four-channel patterns of mean spike waveforms between spontaneous and light-evoked spikes plotted against the energy of light-evoked responses for each neuron. **d** Responses of optogenetically identified iSPNs of DMS and DLS aligned with movement onset and outcome (reward or no-reward) tones. The yellow horizontal line represents a significant difference (p < 0.01, bin = 20 ms; t-test) between rewarded (red) and non-rewarded (blue) responses. Horizontal black bars indicate outcome tone duration. **e** Reward rate correlation of DMS-iSPNs with Go cue period activity during rewarded (red) and non-rewarded (blue) trials. All error bars represent SEM. On each box, the central dot indicates the median, the bottom and top edges of the box indicate the 25th and 75th percentiles, respectively. The whiskers show the extreme data points not considered outliers. The histograms show r value distribution of individual neurons; colored bars represent significant correlation. The central mark on the box plot indicates the median, the bottom and top edges of the box indicate the 25th and 75th percentiles, respectively. The whiskers show the extreme data points not considered outliers; p value indicates the comparison between the distribution of r values and 0 (Wilcoxon rank-sum test). **f** Same as **e** for movement onset period. **g** Same as **e** for outcome period. **h** Reward rate correlation of DLS-iSPNs during Go cue period. **i** Same as **h** for movement onset period. **j** Same as **h** for outcome period. **k** Comparison of the strength of the neural activity correlation with reward rate among DMS and DLS SPNs. z-score of the correlation coefficients using Fisher's z-transformation. Each dot represents one neuron. The error bars correspond to the confidence interval; *p < 0.05, **p < 0.01; α = 0.05. Fisher's z-test. The p-values were corrected for multiple comparisons using Bonferroni method.

through dopamine D2 receptors[13]. It is important to note that the modulation of iSPN activity by dopamine is not proposed to occur instantaneously but rather in correlation with the patterns of dopamine release over a broader time window. As seen in our results, the onset of dopamine release and changes in SNc DAN and iSPN activity do not align on a moment-to-moment basis, suggesting the impact of other factors in this complex dynamic. Dopamine effects can be multifaceted, impacting different aspects of neuronal function including excitability, synaptic plasticity, and long-term changes in gene expression, which are context dependent. It is worth emphasizing that our study does not imply a direct influence of dopamine on iSPN activity, but rather

highlights the role of reward expectation in modulating neural activity, potentially through other parallel inputs. As evidenced by our observations, the precise temporal dynamics of this interplay remain an intriguing area for future investigation.

The processing of reward information, as manifested in the action-related activity, might be under the influence of local striatal control, as well as be affected by inputs other than dopamine (Fig. 6). This includes inputs from diverse brain regions such as the medial prefrontal cortex, anterior cingulate cortex, orbitofrontal cortex, motor cortex, thalamus, and globus pallidus. In particular, the role of intratelencephalic (IT) neurons, which constitute an important portion of the corticostriatal

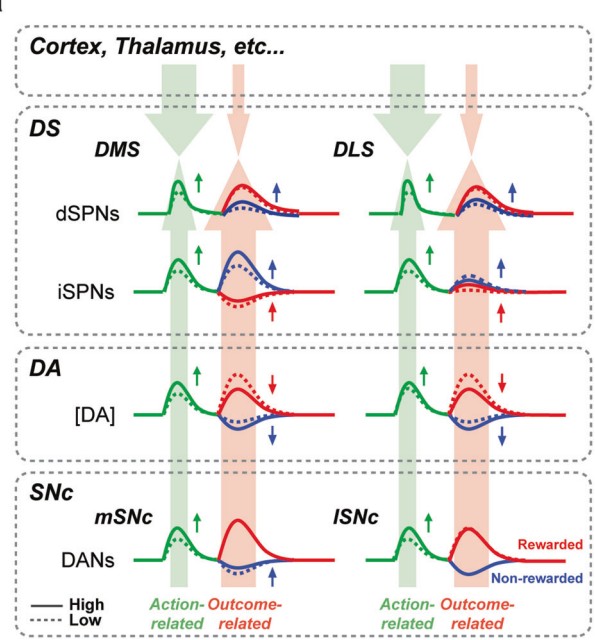

**Fig. 6 Activity modulation of nigrostriatal systems by recent rewards.**
**a** The action-related activity (Go cue and movement onset; green lines) exhibited a homogeneous enhancement (higher firing activity or local DA release) with higher reward rate (solid line), i.e., reward expectation, across diverse neuron types and regions. Contrastingly, the outcome-related activity (red lines, rewarded; blue lines, non-rewarded) showed a heterogeneous relationship with reward rate. Small vertical arrows indicate activity increase/decrease with higher reward rate. Large light green and red arrows represent possible effects of reward expectation on action- and outcome-related activities, respectively.

projections, cannot be overlooked. These IT neurons, which send projections from the cortex to the striatum, are known to play a critical role in information processing within the striatum, influencing the activity of SPNs during tasks such as the one we used in the present study. There is a large body of evidence suggesting the central role of IT neurons in modulating striatal activity, emphasizing their importance in our understanding of reward-related neuronal activity[46,47]. In addition, it has been observed that pyramidal tract-type neurons in the motor cortex provide collateral input to the dorsolateral striatum. This input may convey specific or biased information to the downstream striatal circuits[48]. This illustrates the complexity and multifaceted nature of inputs into the striatum, and the variety of potential sources that could influence the activity of SPNs in a task-dependent manner.

Furthermore, we evaluated the reward information processing in medial and lateral nigrostriatal systems. In rodents, the SNc and dorsal striatum are thought to have medial and lateral topographical and reciprocal connections reflecting different functions[4,49]; for example, mSNc-DANs encode signals related to motivational value, whereas lSNc-DANs responsible for motivationally related salience[4,26]. Likewise, DMS is involved in goal-directed behavior whereas DLS is involved in habitual behavior[33–35]. Nonetheless, our results suggest that during choice selection, nigral and striatal neuron systems convey similar reward information regardless of their medio-lateral topographical segregation, in agreement with dopamine signal restoration studies showing that many cognitive processes are shared by DMS and DLS[50,51]. Additionally, we observed a substantial number of neurons whose Go cue- and movement-related activity correlated with reaction time, which may suggest that the

activation during this period is monitoring movement dynamics. Among all the identified populations, the DLS-SNc and DMS-SNc DANs had the higher proportion of units correlated with reaction time (Supplementary Fig. 8), which may suggest that the activation during this period is monitoring movement dynamics in addition to reward monitoring.

On the other hand, the outcome-related activity, typically differentiable according to reward or no-reward, was distinctly modified by recent rewards, depending on the outcome situation, neuron subtype, and area (Fig. 6, red and blue arrows). This discrepancy during the action and outcome periods suggest that reward rate may be processed in two different modes, action-related and outcome-related modes, by the same set of basal ganglia neurons. Moreover, during the outcome period, only a small number of units exhibited a correlation with reaction time or movement time (Supplementary Fig. 8) suggesting that contrary to the movement-related activity, the activity during outcome period is not related to monitoring movement dynamics, but with monitoring the reward or no-reward of selected action. The uniformity of the correlation pattern of movement period with reward rate could reflect an increase in the value of the external cue or actions, or an increase in motivation[37,41,52], whereas the diversity during the outcome period may result from adaptive coordination for different brain functions such as outcome feedback, action selection[18], state change[53] and motivation[54]. Movement-related activity in the dorsal striatum as well as SNc may be driven by robust inputs including the cortical and thalamic inputs[43,55], rendering it less susceptible to modulation by dopamine inputs. Still, the SPNs can be affected by dopamine from SNc DANs activated simultaneously (Fig. 6 and Supplementary Fig. 7). Previous studies suggest that phasic dopamine release in the dorsal striatum is crucial for regulating movement intensity, where increased motivation is linked to increased actions to obtain a reward[56]. Our findings echo this relationship, as we identified a similar interplay between movement-associated activity in DMS-SNc neurons and SPNs, hinting at a plausible link between dopamine levels and movement intensity under our experimental context. Also, the latency of the DANs firing activity was shorter than that of the dopamine signal response, as expected considering the time needed for dopamine release and uptake[39]. As for the SNc DANs, the reinforcement learning theory can consistently explain their enhanced action-related activity and reduced outcome-related activity as encoding of an RPE or temporal difference (TD) error[19]. This pattern indicates that the basal ganglia system may dynamically alter its integration mode, facilitating the real-time processing of past outcome information with each instance of action-outcome feedback.

Additionally, contrary to the movement period, during the outcome period (when the result of the action was presented), the identified DMS-iSPNs responded strongly to the no-reward outcome (Fig. 5d)[18], while DMS-projecting SNc-DANs activity and dopamine signal decreased (Figs. 2b and 3c). These results indicate that the effect exerted by the dopamine on the DMS during the outcome period agrees with the classical models, exciting dSPN and suppressing iSPN. Nevertheless, we observed some discrepancies between the firing activity of DANs and the dopamine release measured by fiber photometry. In contrast to the DMS-projecting SNc-DAN responses to no reward (Fig. 3f, i), the outcome-related dopamine signal was higher when reward expectation was lower, consistent with the RPE (Fig. 2f, i). Discrepancies between DAN activity in the ventral tegmental area and dopamine release in the nucleus accumbens have also been reported in behaving rodents[37]. Another potential explanation is that the influence of the reward rate on dopamine release during the feedback period may be locally controlled. Previous studies

revealed that TANs and FSIs play important roles in the processing of reward information[57–59]. For example, it was reported that cholinergic interneurons (corresponding to TANs) in the striatum may accomplish control of striatal dopamine signaling independent of the activity of midbrain cell bodies[60]. In our study, action-related activity of TANs and FSIs was positively correlated with reward rate, like other neuron types. Also, although the outcome response was subtle, putative DMS-TANs and FSIs exhibited a pattern influenced by the reward rate (Fig. 6).

In further studies, more precise examination of the activity of various neuron types with consideration of pathway-specific activity is essential for advancing our understanding of how reward and action information is integrated in the nigral and striatal neuron systems. Additionally, although there are some limitations in our study addressing a direct dependence of the neural activity on reward history, and further investigations that manipulate reward expectation directly are necessary, this study provides insights into the neural mechanisms underlying reward processing and highlights the potential role of reward expectation in modulating neural activity, shedding light on the neural mechanisms underlying the link between reward and motor control. In conclusion, we have defined the action-related activity of diverse nigral and striatal neuron types is positively enhanced by reward rate, providing further insights into the mechanism of the basal ganglia circuitry underlying reward-based action selection and execution.

## Methods

**Animals and surgery**. All experiments were approved by the Animal Care and Use Committee of Tokyo Medical and Dental University (A2019-274, A2021-041), and were performed in accordance with the Fundamental Guidelines for Proper Conduct of Animal Experiment and Related Activities in Academic Research Institutions (MEXT, Japan) and the Guidelines for Animal Experimentation in Neuroscience (Japan Neuroscience Society). All surgical procedures were performed under appropriate isoflurane anesthesia, and all efforts were made to minimize suffering. Our procedures for animal experiments are described in our previous studies[18,43].

Seven tachykinin precursor 1 (Tac1-Cre) transgenic rats (246 ± 28 g, all male), six dopamine D2 receptor-Cre (Drd2-Cre) transgenic rats (262 ± 24 g, all male), seven tyrosine hydroxylase-Cre (TH-Cre) transgenic rats (269 ± 38 g, all male), and six wild-type (WT) Long-Evans rats (248 ± 31 g, all male) were kept in their home cages under an inverted light schedule (lights off at 9:00 A.M.; lights on at 9:00 P.M.). The rats were briefly handled by the experimenter (10 min, twice) in advance. For head-plate (CFR-2, Narishige) implantation, the animals were anesthetized with isoflurane (4.5% for induction and 2.0–2.5% for maintenance, Pfizer) using an inhalation anesthesia apparatus (Univentor 400 anesthesia unit, Univentor) and were placed on a stereotaxic frame (SR-10R-HT, Narishige). Lidocaine (Astra Zeneca) was administered around the surgical incisions. Reference and ground electrodes (Teflon-coated silver wires, A-M Systems; 125 µm diameter) were implanted above the cerebellum. During anesthesia, body temperature was maintained at 37 °C using an animal warmer (BWT-100, Bio Research Center). Analgesics and antibiotics were applied postoperatively, as required (meloxicam, 1 mg/kg s.c., Boehringer Ingelheim; gentamicin ointment, 0.1%, MSD).

After full recovery from surgery (3–7 days later), the rats had ad libitum access to water during the weekends, but during the rest of the week they obtained water only when they performed the task correctly. When necessary, agar was given to the rats in their home cage to maintain them at >85% of original body weight[55].

**Vector injection**. For the optogenetic identification and manipulation experiments, 1 µl of AAV2-EF1α-Flex-ChRWR/Venus (titer: $2.06 \times 10^{12}$ vg/mL) or AAV2-Syn-Flex-rcChrimsonR-tdTomato (titer: $8.32 \times 10^{12}$ vg/ml) vector was injected into the DMS (A: +1.0, L: +2.3, DV: +3.7 mm from bregma), DLS (A: −1.0, L: +4.5, DV: +4.6), or SNc (A: −5.5, L: +1.8, DV: +7.5). For the fiber photometry experiments, 0.5 µl of AAV9-Syn-dLight1.3b ($1 \times 10^{13}$ vg/mL; Addgene, MA, USA) was injected into the DMS (A: +1.0, L: +2.3, DV: +3.7) or DLS (A: −1.0, L: +4.5, DV: +4.6). The vector solution was slowly injected (200 nl/min) through a 50 µm tip glass capillary using a syringe pump (Legato 100; KD Scientific) with a 10 µl Hamilton syringe. After the injection, the capillary was left in place for 10 min. The experiments were performed at least 3 weeks after the injections.

**Behavioral choice task for probabilistic reward**. The push/pull choice task based on probabilistic reward was performed by head-fixed rats in a sound-attenuated chamber shielded from light and electricity (60 × 75 × 60 cm, custom-made by O'Hara & Co.). The rats obtained water as a reward (5 µl, three times in 0.3 s, dispensed by a micropump) from a spout in front of their mouth. Orofacial movements related to the consumption of the reward water were monitored by measuring the torque on the spout (KFG-2N amplified by DPM-711B; Kyowa). On the first day of training, the rats were rewarded immediately after they grasped a lever and after they either pushed it forward or pulled it back. The lever glided over a rail of the acrylic plate and returned automatically toward the center position if released. The rats adjusted the lever to the center position to start the next trial. Stoppers for push and pull targets were separated by 10 degrees. On consecutive training days, after the animals were able to achieve rewards 300–500 times, the lever push or pull was followed by a high-tone sound (instructing a reward outcome; 10 kHz, 60 dB, 0.3 s), whereas the opposite movement was followed by a low-tone sound (instructing a no-reward outcome; 4 kHz, 60 dB, 0.3 s). The 10 kHz tone was followed by delivery of reward water with a delay of 0.3 s, whereas the 4 kHz tone was not followed by a reward. Over the course of 3 days, most rats learned to choose the reward action consistently after about ten trials. Next, before pushing or pulling, the rats needed to hold the lever at the center position for an incremental duration of up to 0.4 ± 0.1 s (Hold 1, pre-Go, Fig. 1a), and then push or pull the lever in response to the Go signal (chamber light, 30 lux). After reaching the push or pull target, rats were required to keep holding the lever for 0.3 s. The Go light stayed on until 1 s after the reward or no-reward tone onset but turned off when the rats erroneously moved or released the lever. As a final step, the high- and low-tone sounds indicating reward and no reward, respectively, became probabilistic for each movement (rewarded at 70% for push, and at 10% for pull in Fig. 1a). There were two blocks of trials with different action–outcome contingencies (push block: push 70% versus pull 10%; pull block: push 10% versus pull 70%), and these were alternated when the rats selected the high-value option in more than 79% of the previous ten trials (about 30–50 trials). Task training and recording of neuronal and behavioral data continued for 3–4 h a day (total: at least 15 training days in 3 weeks). The lever position was always monitored by the task-training system through an 8-bit encoder.

**Behavioral analysis**. The reaction time was defined as the point where the cumulative sum of the lever trajectory drifts more than three standard deviations beyond mean trajectory during the

holding period. The movement time was defined as the time from movement onset until the lever entered the pull or push area. The contributions of past rewards or lack of rewards on the animals' current choice were analyzed on a trial-by-trial basis using the following logistic regression model:

$$\ln\left(\frac{P_S(i)}{1 - P_S(i)}\right) = \sum_{j=1}^{n}\left(\beta_j \text{Reward}\right) + \sum_{j=1}^{n}\left(\beta_j \text{No Reward}\right) + \beta_0 \tag{1}$$

where $P_S(i)$ is the probability of push selection in the $i$th trial, and n indicates the number of past trials included in the model ($n = 8$). The regression coefficients $\beta_j^{\text{Reward}}$ and $\beta_j^{\text{No-Reward}}$ represent the contributions of past rewards and lack of rewards, respectively, while $\beta_0$ indicates the intrinsic bias of the animal.

**Fiber photometry**. The genetically encoded optical dopamine sensor dLight 1.3 was expressed in dorsal striatum using a viral vector approach. In the same surgery, a fiber optic cannula (200 μm core, Thorlabs) was inserted (~100 μm in-depth dorsal to the AAV injection site) and fixed using dental cement. For signal acquisition, custom-written LabVIEW software was modified to control the recording hardware. LEDs of 470 nm (for dLight) and 405 nm (for isosbestic signal) were alternately turned on and off at 40 Hz, and bulk fluorescence was acquired using a photo-multiplier (H10721; Hamamatsu, photonics). The signal was sampled at 1 kHz with a data acquisition device (NI USB 6211) and downsampled to 40 Hz for further analysis. The acquired photometry data were processed with custom-written code in MATLAB. First, a fitting curve was estimated and subtracted from the original signal to remove exponential and linear signal decay during the recording session. A linear fit was applied to align the 405 nm signal to the 470 nm signal, and then the fitted 405 nm signal was subtracted from the 470 nm channel and divided by the fitted 405 nm signal to calculate $\Delta F/F$ values. The $\Delta F/F$ time-series trace was normalized using z-scores to account for data variability across animals and sessions[61,62]. Representative 470 nm and 405 nm signals are shown separately in Fig. 2b.

**Electrophysiological recording**. Supported by agarose gel (2% agarose-HGT, Nacalai Tesque) on the brain, a 64-channel silicon probe (Isomura64-4x-tet-lin-A64, with 16 tetrode-like arrangements on four shanks; NeuroNexus Technologies) was inserted vertically into the left DMS (A: +1.0, L: +2.3, DV: +3.7 mm from bregma), DLS (A: −1, L: 4.5, DV: 4.6 from bregma), or SNc (A: −5.5, L: 0.8 ~ 2.4, DV: 7.5 from bregma) using a micro-manipulator (SM-15A, Narishige) on a stereotaxic frame (SR-10R-HT, Narishige). Multineuronal (multiple isolated single unit) recordings were performed during the conditioning task. The neuronal activity was amplified by two main amplifiers (32 channel: FA-32, Multi-Channel Systems; final gain, 2000; band-pass filter, 0.5 Hz to 10 kHz) through two 32-channel head-stage miniature preamplifiers (MPA 32I, Multi-Channel Systems; gain 10). The amplified signals were digitized at 20 kHz with two 32-channel hard-disk recorders (LX-120, TEAC).

**Spike isolation**. Raw signal data were processed offline to isolate the spike events of individual neurons in each tetrode. Spike candidates were detected and clustered using EToS semiautomatic spike-sorting software[63,64]. Using the manual clustering software Klusters and the viewing software NeuroScope[65], spike clusters were further combined, divided, and/or discarded manually, to refine single-neuron clusters according to the presence of refractory periods (<2 ms) in the autocorrelograms and the absence of refractory periods in cross-correlograms with other clusters.

**Optogenetic identification of DANs and iSPNs**. After completing a session of recording neuronal activity during the operant conditioning task, we investigated whether the recorded neurons were responsive to light stimulation. To identify striatum-projecting SNc neurons, we used the light-evoked collision test, which was previously established[18,43,44,55] (see Fig. 3b and Supplementary Fig. 5). Briefly, before the insertion of silicon probes, an optical fiber for stimulation was placed on either DMS or DLS using a micromanipulator (SM-25A; Narishige). To evoke antidromic spikes in specific axonal projections, a blue LED light pulse (intensity, 5–10 mW; duration, 0.5–2 ms, typically 1 ms) was applied through each of the two optical fibers using an ultra-high-power LED light source (see below) and a stimulator. To be classified as projecting neurons, neurons were required to meet several criteria, including constant latency, high frequency following (frequency-following test, two pulses at 100–250 Hz), and collision tests[66]. The collision test was visually confirmed online to accumulate spike collision data that would be sufficient for post hoc analysis. After offline sorting for spike isolation, we compared filtered tetrode (four-channel) traces that had no spikes before the stimulus (see Fig. 3b, black, control traces) with those that had a spike in one spike cluster (see Fig. 3b, yellow test traces). MATLAB (The MathWorks) was used for these comparisons. If we found antidromic-like (all-or-none and no jittering) spike activities with constant latency (<25 ms) in many of the control traces (Supplementary Fig. 5a, b), we set a time window for counting possible antidromic spikes based on a clear dissociation between averaged control and test traces due to the presence or absence of spikes. A cutoff threshold defined from a receiver operating characteristic (ROC) curve for the distribution of the most negative points within the time window was used to determine whether the spikes were present or not, allowing us to obtain spike and no-spike counts in the control and test events. According to this method, we included spike clusters with control spike probability >50% and test spike probability less than half that of the control. Finally, the passing of the collision test was statistically justified by a $2 \times 2$ chi-square test ($p < 0.05$) for spike and no-spike counts in control and test events. The latency of antidromic spikes was defined as the time from the onset of stimulation to the median (the second quartile, 50%) of their peak positions within the time window, and their jitter was defined as the time between the first (25%) and third (75%) quartiles of their peak positions within the time window. In this way, we judged these spikes to be antidromic or not based on the collisional disappearance of antidromic spikes (collision test), as well as their all-or-none properties, absence of jitter (constant latency test; <0.5 ms), and high reliability (frequency-following test; if applicable in the tentative collision test).

To identify iSPNs, an optical fiber was inserted into the external segment of the globus pallidus (GPe; A: −2.0, L: 3.0, DV: 5.5) in six Drd2-Cre rats to stimulate axons of ChR- or ChrimsonR-expressing striatopallidal neurons. A single light pulse (460 nm or 595 nm, <10 mW, 1.0 ms duration) was delivered through the optical fiber (FT400EMT, FC, Thorlabs; NA, 0.39; internal/external diameter, 400/425 μm). The light pulses were generated by an ultra-high-power LED light source (UHP-Mic-LED-460/595, FC; Prizmatix) triggered by a stimulator (SEN-8203; Nihon Kohden) every 3 s[18]. We considered action potentials to have been elicited by the light pulse if they occurred at a latency shorter than 10 ms with short jittering (<1 ms). We calculated the correlation coefficient for four-channel patterns of mean spike waveforms (0.125 ms before and after the spike peak) between spontaneous and light-evoked spikes and plotted it against the energy of light-evoked responses for each neuron. The light-evoked spike waveforms needed to appear almost identical (correlation coefficient >0.9) to

spontaneously occurring waveforms. For the dSPNs identification, a single light pulse was delivered through the optical fiber in the DMS or DLS (adjacent to recording sites) in Tac1-Cre rats to stimulate axons of ChR- or ChrimsonR-expressing striatonigral neurons. We considered action potentials to have been elicited by the light pulse if they occurred at a shorter latency than 8 ms with small jittering (<1 ms). We performed similar waveform correlation analysis as for iSPNs identification (Figs. 4b, c and 5b, c).

**Selection of putative SNc and striatal neurons.** For the selection of putative SNc DANs and GABAergic interneurons, the spike duration of the SNc recorded neurons was defined as the time from spike onset to the first positive peak of the aligned averaged spike waveform. Different subpopulations formed two clearly separable clusters. Non-optogenetically identified SNc neurons with a spike duration <800 μs were classified as non-dopamine cells. We further subdivided putative DMS- or DLS-projecting neurons according to their mediolateral recording location (medial: 0.8–1.8 mm; lateral: 1.8–2.4 mm from bregma). To ensure that recorded neurons were located in the SNc, we only analyzed cells recorded during sessions with at least one optically identified dopamine cell.

Striatal nonoptically identified task-related neurons were further subdivided into putative SPNs, TANs, or FSIs using k-means clustering applied to three parameters: 1) ongoing spike rate, 2) peak-to-valley width of the four-channel average spike waveform, and 3) coefficient of variation (CV) of the interspike interval (ISI). The different subpopulations formed three clearly separable clusters[67] (Supplementary Fig. 6a, b). The silhouette technique was used to evaluate the cluster separation (average silhouette value = 0.603). The waveform peak-to-valley widths were calculated using the waveforms of the whole recording periods. SPN waveforms were typically wide and displayed a long positive second phase. SPNs and TANs were further subdivided into putative DMS- and DLS-SPNs, and DMS- and DLS-TANs, according to the recording site. We did not perform further subdivisions of FSI neurons because of their small numbers.

**Immunofluorescence.** Brains were transcardially fixed with 4% paraformaldehyde/0.1 M phosphate buffer under deep anesthesia and sectioned with a vibratome (VT1000S, Leica, Wetzlar, Germany). Brain sections were incubated with specific primary antibody, namely, polyclonal anti-GFP (RRID: AB_221569; 1:1000), polyclonal anti-RFP (AB_10013483; 0.25 mg/ml, 1:1000), or polyclonal anti-TH (AB_741693; 1 mg/ml, 1:1000), followed by species-specific secondary antibodies conjugated with Alexa Fluor 488 (AB_2556542; 2 mg/ml, 1:500) or Alexa Fluor 555 (AB_2762834; 2 mg/ml, 1:500; Invitrogen, ThermoFisher). Images were acquired on a confocal microscope (FV1000, Olympus, Tokyo, Japan).

**Statistics and reproducibility.** To determine whether a neuron showed a significant action-related and/or outcome-related response, we applied the Wilcoxon rank–sum test to trial-by-trial firing rates during the baseline period (2–3 s before the Go-cue), 0.3 s period after the Go-cue (Go window), 0.3 s period after movement onset (movement window), and 0.5 s period after the onset of reward and no-reward tones (outcome tone window). The perievent time histograms (PETH; bin width, 20 ms) were smoothed by averaging across a 20 ms sliding window with a 10 ms step. Rewarded (red) and non-rewarded (blue) responses were compared using Student's *t* test. Because of the considerable variation in background firing rates of individual neurons, we used z-score normalization to measure neuronal responses. To measure responses to movement onset and outcomes, the

discharge rates from a given task period were adjusted by subtracting the mean firing rate of the baseline period and then dividing the difference by the standard deviation of the baseline period. After selecting the task-related neurons using the above-described criteria, we determined whether the firing rate was preferentially related to a specific action (push or pull) by comparing the mean activity after pushing or pulling during the movement window. For this further analysis, we selected only neurons that had a spike rate difference of at least 5% between push and pull trials. We only considered the preferred action activity for the reward rate analysis. To quantify the degree of recently obtained rewards, the reward rate was calculated as the mean number of rewards received in the last five trials (without including current trial), i.e., the number of rewards divided by five (reward = 1; no-reward = 0), irrespective of the selection. This allowed us to quantify the average rewards received recently, providing a window into the animal's immediate reward history. We obtain the Pearson's correlation coefficient to investigate the relationship between the reward rate and neuronal activity (expressed as a z-score) in individual neurons. We plotted these correlations, with the reward rate on the x-axis and the corresponding z-scored neuronal activity on the y-axis. The slope of the lines in these plots represents the strength and direction of the correlation; positive slopes indicate that neuronal activity increases with higher reward rate, while negative slopes suggest the activity decreases with higher reward rate. We assessed the distribution of correlation coefficient ($r$) for a particular neuronal population at a specific task event. And compared the distribution of $r$ values and 0 using Wilcoxon rank-sum test (Figs. 2–5). Additionally, we visualized a tendency of the distribution of correlation coefficients in a boxplot for each dataset. On each box plot, the central mark indicates the median, the bottom and top edges of the box indicate the 25th and 75th percentiles, respectively. The whiskers show the extreme data points not considered outliers. Supplementary Table 1 shows statistical tests of dLight signal and neuron types. To compare the strength of the correlation between two or more populations, we used the Fisher's z-test of the correlation coefficients with Bonferroni correction for multiple comparison.

**Reporting summary.** Further information on research design is available in the Nature Portfolio Reporting Summary linked to this article.

## Data availability

Data needed to evaluate the conclusions in the paper are present in the paper and/or the supplementary materials. Source data underlying figures can be consulted from: https://doi.org/10.5281/zenodo.8282962.

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

## Acknowledgements
This work was supported by Brain/MINDS (JP19dm0207089 to Y.I.; JP18dm0307005 and JP21dm0207115 to A.N.; JP22dm0207113 to K.K) from AMED; by CREST (JPMJCR1751 to Y.I.) from JST; by Grants-in-Aid for Scientific Research on Innovative Areas (JP16H06276 and JP20H05053 to Y.I.; JP18H05524 to Y.S.) and Transformative Research Areas (A) (JP21H05242 to Y.I; JP22H05497 to S.N.; JP23H04688 to A.N.) from MEXT; by Scientific Research (B) (JP19H03342 and JP23H02589 to Y.I.; JP23H02594 to A.N.); Fund for the Promotion of Joint International Research (B) (JP19KK0193 to A.N.) and Early-Career Scientists (JP21K15184 to A.R.; JP19K16300 and JP22K15226 to S.N.) from JSPS; and by the Takeda Science Foundation (Y.I.).

## Author contributions
Conceptualization: A.R., S.N., Y.S., M.K., and Y.I.; Methodology: A.R., S.N., S.K., J.Y., N.M, R.H., Y.S., M.K., and Y.I.; Investigation: A.R. and S.N.; Visualization: A.R. and S.N.; Supervision: A.N., M.T., K.K., Y.S., M.K., and Y.I.; Writing—original draft: A.R., S.N., and Y.I.; Writing—review & editing: A.R., S.N., and Y.I.

## Competing interests
The authors declare no competing interests.
