## [Peer Review File · Communications Biology]

Reviewers' comments:

Reviewer #1 (Remarks to the Author):

Review of

Reward expectation enhances action-related activity of striatal indirect pathway 3 neurons regardless of elevated nigral dopamine input

In the Abstract and in the main text: clarification is needed of how "action-related activity", "reward-expectation", are defined and distinct from "outcome-related" activity

The authors used a behavioral task that demonstrated that "reward experience" determined the "action choice" selected by the animal.

- Reward experience is referred to also as reward rate. This is confusing throughout as these different terms are used at various places and difficult to know if they are intended to refer to different things.

They then determined the correlation between dopamine release in the striatum during different parts of the reward choice task using dLight expressed in the DMS or DLS striatum.

- In both DMS and DLS DA release increased after the Go-cue with an ascending phase coincident with movement onset independent of outcome.
- In both DMS and DLS there was also an increase after reward and a decrease when there was no reward.
- Difference in time course with DA response rapidly increasing and back to baseline in DLS while slower and sustained in DMS.

As it had been shown before that DA release in the striatum is not always correlated with firing patterns of SNc DA neurons, activity of DA and GABAergic neurons in the SNc was analyzed during the task. DMS and DLS projecting SNc DA neurons were compared.

- Both DMS and DLS robust phasic after Go-cue
- Outcome activity: increased with reward / decreased with no-reward
- DLS suppression less than DMS after no reward
- After Movement onset reward rate: both DMS and DLS SNc positively correlated
- Outcome: only SNc-DMS positively correlated SNc-DLS negatively correlated

Clarification needed of: " Additionally, the reward experience dependence of the outcome-related activity of SNc-DMS DAN notably differed from the d-Light signal in the DMS." Some explanation of this statement is required, what specific data is being compared and how do they differ?

A question: in Figure 3g, is it correct that the animal doesn't know at the time of the "Go-cue" and "Movement onset" whether they will be rewarded? If so is the difference between Rewarded and Non-rewarded at the time of the Go-cue based on prior experience or is there a difference, some explanation is required.

Relationship of activity of dSPNs and iSPNs in the DLS and DMS during the reward task were analyzed.

The study does not actually demonstrate the "Dependence of dSPNs and iSPNs on reward expectation" at most they show a correlation

The results described need more clarification.

- what is the "action period" is it the time after the GoCue, or the "movement onset" period or "outcome tone" period? This needs to be clarified to understand the conclusions reported particularly in relationship to how they use the data to refer to "reward experience" very confusing, when different terms are used
- would be useful to describe in Figures 4 and 5 what information is being provided in plots c and d in language that doesn't require knowing what z score and r value are determined and what is different

between what "c" and "d" provides in terms of information

- dSPNs they report "higher activation in response to rewarded than non-rewarded outcomes", what about during the "movement" onset period,
- is the difference between the rewarded and non-rewarded considered to be "weak" for these examples
- The statement that "only action- and outcome- related activity of DLS-dSPNs during non-rewarded trials were positively correlated to reward rate" needs to be explained in terms of what data they are using specifically to make this statement. – is this based on the histogram data in fig 4 "d" if so, need to provide more description
- The results for the iSPNs need clarification: for DMS-iSPNs, in C) the response is significantly greater to the non-rewarded outcome tone than the rewarded outcome tone, but in D) looks like it is the opposite
- As exemplified by the above the difficulty of understanding what the data in Figures 4 and 5 are intended to demonstrate a better description of the experimental paradigm is required- exactly how is "reward experience" determined

In addition to comments above there are other problems with this study.

1. after multiple readings it was possible to figure out that the results of the "activity" in dSPNs and iSPNs during the task was presented in two ways, one was the actual "activity" of the SPNs and the other was a measure of the "activity" relative to the "reward expectation/reward rate". This needs to be clarified, both in the set-up of the experiments, and in the description of the results.
2. The authors try to make the case that since DA release in the striatum appears to be correlated with "reward expectation" that the value of the "activity measured relative to reward expectation/rate" is "dependent" on striatal DA release. While they might argue that they are correlated, this study does not provide evidence of "dependence".
3. The authors are aware of this as they state "The reward information processing reflected in the action-related activity may be under local striatal control and/or control from input other than dopamine." There are other possible sources not mentioned, including the thalamus and the globus pallidus in addition to the cortex, which is mentioned.
4. The authors use the term "modulated" in 5 instances without providing evidence.
5. In the discussion there are several concerns with this sentence: "Indeed, the action-related activities of the identified DMS- and DLS-iSPNs were positively modulated by the reward experience, suggesting that reward experience-dependent action information is conserved throughout the mediolateral nigrostriatal system, mainly throughout the striato-pallidal pathway (Fig. 5)."
 - o "modulated by" needs to be changed to "correlated with"
 - o Throughout the paper it is confusing "action related activities" are the same as "reward experience", in this sentence they are used as though they are the same, but in other parts of the studies it seemed as though a distinction is made (such as in Figure 5C and 5D).
 - o In Figure 5C and 5D it appeared that there are differences for iSPNs between the DMS and DLS, but here it is suggested they are not different, clarification is needed.

In the Abstract and in the main text: clarification is needed of how "action-related activity", "reward-expectation", are defined and distinct "outcome-related" activity

A major problem is the use of a "strawman" to contrast their findings with their view of a "classical model". This is an overused construct in an attempt to claim experimental findings are novel. What specifically are the author's referring to as the "classical model". An example of the classic model of the basal ganglia is that the direct and indirect pathways originate from separate populations of SPNs, which respectively express the D1 and D2 dopamine receptors and that activity in the direct pathway promotes and activity in the indirect pathway suppresses movements. This "model" was originally proposed based on clinical disorders, such as Parkinson's disease, where the absence of DA results in activity in the indirect pathway dominating to produce bradykinesia. Very soon after this model was

proposed (and it might be more accurate to describe the original as a model of clinical disorders) Mink (1996) proposed that normal motor behavior involves promotion of selected actions and suppression of competitive actions respectively by activity in the direct and indirect pathways. In keeping with this concept, numerous studies have shown that both dSPNs and iSPNs are active in a wide range of behavioral paradigms. As this conceptual framework is refined studies have expanded ideas of what constitutes selected and competing actions to recognize that activity in the direct and indirect pathways act both cooperatively and antagonistically.

Reviewer #2 (Remarks to the Author):

This study analyzed the impact of dopaminergic signals in the activity of the direct and indirect striatal pathways in the context of forelimb movements and its rewarding consequences. Throughout a series of complex experiments combining a probabilistic task with fiber photometry and high-density electrophysiological recordings targeting the substantia nigra pars compacta (SNc) and the dorsolateral (DLS) and dorsomedial (DMS) striatum, the authors evaluated striatal dopamine release and the spiking activity of identified iSPNs, dSPNs, and SNc dopaminergic neurons during movement production and reward. The results may be divided in two streams. First, those related to movement initiation, where average dopaminergic activity (release, and SNc spiking activity), and most relevantly, striatal spiking activity in both, direct and indirect pathways, increased during movement production. These observations challenge the canonical views on function of the basal ganglia (BG) during movement production. And second, those results related to reward consequences, where dopaminergic neurons spiking activity in SNc and dopamine release in dorsal striatum increased after rewarded trials but decreased during non-rewarded ones. In my opinion, the experimental procedures and results are well designed and carefully performed. The observations may be of great relevance for our understanding of the BG, however, I still have some questions regarding the presentation and interpretation of these results.

Major comments

1. I believe that the title of the MS does not entirely reflect the general findings of the work. It is completely biased to the observations related to reward expectation and indirect pathway. But in my opinion, another fundamental finding of this work is that, around movement initiation, there is a general dopaminergic-related cascade of events that includes an increase in dopaminergic release in the dorsal striatum and an increase in spiking activity in SNc spiking and both main striatal subpopulations iSPNs, dSPNs. While the specific consequences of this generalized activation will need to be thoroughly explored, the observations are at odds with what is expected at the onset of movements from the classical model of the BG. In my opinion, these findings challenge validity of the general model of the BG during movement initiation.

2. Reward rate should be thoroughly explained in the results or methods section. I found that the general concept of Reward Rate is not entirely clear and required two or three times for the reader to finally grasp the idea that values closer to 0 means that the animals express a higher probability to switch behavior on the next trial, while values closer to 1 means that the animals tend to repeat the same behavior. I believe that clarifying this point is fundamental for the general understanding of the MS.

3. Is interesting that the activation latency of iSPNs and dSPNs are different at movement onset (Fig. 5 & 6C). Was this analyzed? What would be the reason for this difference? On the same line of thinking, dopaminergic spiking activity (Fig. 3) and dopaminergic release (Fig. 2) expressed a clear difference in latencies immediately after reward outcome (Fig. 2C & 3C). However, iSPNs/dSPNs latencies at reward outcome appear to be faster than dopaminergic release (Fig. 4C & 5C). What would be the reason for this apparent discrepancy? I believe that a systematic analysis of response

latencies at movement onset and reward outcome for all experiments would help to clarify the dopaminergic dynamics on this behavioral context.

4. In general, for the electrophysiological experiments, I found that the target populations ended up with significantly lower numbers than the total recorded neurons. For example, in SNc, authors reported more than 1500 neurons recorded, but the dopaminergic neurons included in the final analysis were only 56 (33 and 23 projecting to DMS and DLS, respectively; lines 167-168), that would be about 3.7 % of total recorded neurons. In the case of dorsal striatal SPNs, please provide specific numbers for each subpopulation, for the clustering clouds reported in SFig.5A, it appears that MSNs are very little. With that said, from the 2002 cells recorded in the striatum, the final numbers included in the analysis for dSPNs and iSPNs were 66 and 42, respectively. In both cases, striatum and SNc, the impact of the final small samples on the interpretation of the data must be discussed.

5. In the same context, I found a bit surprising the lack of neurons decreasing their spiking activity during movement onset, especially in the striatum, where several previous publications suggest a higher response heterogeneity in this region, with neurons encoding kinematic, spatial and temporal parameters of specific movements. The fact that reaction times and velocity were nicely correlated with reward rate (Fig. 1F) can be used to further clarify the role of iSPNs, dSPNs and dopaminergic neurons. For example, what is the correlation of individual neurons in SNc and DLS/DMS with these parameters? Is it possible that dopamine and striatal dynamics are monitoring movement dynamics? I think it would be very useful to see the raster plots of a few representative neurons on each region.

Minor Comments

1. Please indicate final numbers of IN and DAN. Gives the impression that DANs are significantly smaller population than IN (lines 153-156)

2. How many neurons were classified as interneurons also produced antidromic spikes? If none, it would be important to state it (lines 167-168)

3. Please include representative raster plots of neural activity for each condition, this could be included in the main or supplementary figures.

4. I found very useful the results summary displayed in supplementary figure 6, if consistent with the guidelines of the journal, I would suggest including it as last main figure.

Responses to the Reviewer #1's comments

In the Abstract and in the main text: clarification is needed of how “action-related activity”, “reward-expectation”, are defined and distinct from “outcome-related” activity

The authors used a behavioral task that demonstrated that “reward experience” determined the “action choice” selected by the animal.

Major comments

1. Reward experience is referred to also as reward rate. This is confusing throughout as these different terms are used at various places and difficult to know if they are intended to refer to different things.

A. We made a clarification of how we quantified the reward experience using the reward rate (the mean of rewards obtained in the past five trials) (L-140). This clarification was made also in the Methods section (L- 682). Finally, we change the term 'reward experience' for 'reward rate' in the text if we judge it could generate confusion in the reader (L-88, 123, 125, 141, 152, 210, 220, 221, 223, 225, 231, 264, 281, 320, 323, 330, 332, 333, 364, 367, 677, 682, 683).

They then determined the correlation between dopamine release in the striatum during different parts of the reward choice task using dLight expressed in the DMS or DLStriatum.

- In both DMS and DLS DA release increased after the Go-cue with an ascending phase coincident with movement onset independent of outcome.
- In both DMS and DLS there was also an increase after reward and a decrease when there was no reward.
- Difference in time course with DA response rapidly increasing and back to baseline in DLS while slower and sustained in DMS.

As it had been shown before that DA release in the striatum is not always correlated with firing patterns of SNc DA neurons, activity of DA and GABAergic neurons in the SNc was analyzed during the task. DMS and DLS projecting SNc DA neurons were compared.

- Both DMS and DLS robust phasic after Go-cue
- Outcome activity: increased with reward / decreased with no-reward
- DLS suppression less than DMS after no reward
- After Movement onset reward rate: both DMS and DLS SNc positively correlated
- Outcome: only SNc-DMS positively correlated SNc-DLS negatively correlated

2. Clarification needed of: “Additionally, the reward experience dependence of the outcome-related activity of SNc-DMS DAN notably differed from the d-Light signal in the DMS.” Some explanation of this statement is required, what specific data is being compared and how do they differ? (L-211)

A. We clarified how the neural activity of the SNc-DMS DANs differed from the d-Light signal in the DMS. (‘the SNc-DMS DAN exhibited a higher activity as the reward rate was higher, while the d-Light signal exhibited the opposite pattern’) (L-225).

3. A question: in Figure 3g, is it correct that the animal doesn’t know at the time of the “Go-cue” and “Movement onset” whether they will be rewarded? If so is the difference between Rewarded and Non-rewarded at the time of the Go-cue based on prior experience or is there a difference, some explanation is required.

A. In both rewarded and no-rewarded trials, the reward rate dependence was higher in SNc-DMS than SNc-DLS (Fig. 3g, e). The correlation of the reaction time with reward rate suggests that animals can predict reward outcome based on the reward history. However, there was not difference in RT between rewarded (263 +/- 140ms) and no-rewarded (277 +/- 148) trials. Also, no significant difference in the movement-related activity was found between rewarded and no rewarded trials in

SNC-DMS or SNC-DLS. This pattern was similar in other populations. This clarification was added in the main text. (L-218).

Additionally, to further evaluate the difference of the go cue- and movement-related activity between rewarded or no-rewarded trials, first, we compared the peak activity and the peak latency of the Go cue- and movement- related activity. The population data showed no significant difference among any SPN population on the peak activity after Go cue or movement onset (Fig E1a) or peak activity latency (Fig E1b). The DANs Go cue-related peak activity did not show significant difference between rewarded and no-rewarded trials. However, the movement related peak activity exhibited a bias towards the rewarded trials in both DMS - SNC and DLS - SNC. According to our results, the activity around movement onset increased as the reward experience was higher. Therefore, we compared the proportion of different reward rates between rewarded and no-rewarded trials. We observed that the no-rewarded trials had a higher proportion of low reward rates (Fig E1h, I). To correct the effect of the reward rate in the activity and peak latency, we independently compared the rewarded and no-rewarded trials with each reward rate. There was no significant difference between rewarded and no-rewarded trials during go cue-related peak activity, movement-related peak activity and latency (Fig E1i-o).

The study does not actually demonstrate the “Dependence of dSPNs and iSPNs on reward expectation” at most they show a correlation. The results described need more clarification.

A. We appreciate the reviewer's feedback on our study. We agree that our study primarily shows a correlation between neural activity and reward history (measured as reward rate). While we did not directly manipulate reward expectation in our study, we measured the neural activity in response to different reward rates. Specifically, we observed a uniform positive correlation between neural activity

and the reward rate, as indicated by increased movement-related activity in response to high reward expectation.

We acknowledge the limitations of our study in addressing causality and agree that further investigations that manipulate reward expectation directly are necessary. However, the probabilistic nature of our task, allow us to evaluate the neural activity under different levels of reward expectancy. We believe that our study provides insights into the neural mechanisms underlying reward processing and highlights the potential role of reward expectation in modulating neural activity. It is important to note that previous studies have shown that the activity of dopaminergic neurons is strongly modulated by reward expectation (Schultz, 2016; Bromberg-Martin et al., 2010). Similarly, a previous study (Hikosaka, 2000) showed that both OFC and LPFC activity were sensitive to the expectancy of delivery of a reward. Our study builds on this previous work and extends it by showing that the movement-related activity of SPN and SNc DAergic neurons differs depending on the reward expectation. We believe this is an important contribution to the field as it sheds light on the neural mechanisms underlying the link between reward and motor control.

Overall, we agree that further clarification is needed to fully address the relationship between neural activity and reward expectation in our study.

We comment this issue in the 'Discussion' (L-421).

4. What is the "action period" (clarify each time we mention the action period if it refers to go cue or movement) is it the time after the GoCue, or the "movement onset" period or "outcome tone" period? This needs to be clarified to understand the conclusions reported particularly in relationship to how they use the data to refer to "reward experience" very confusing, when different terms are used.

A. Due to the short interval between the go cue and the movement onset (average < 300ms), we conceptualized the action period as the time around movement onset (where some overlap from the response to the go-cue may occur in short reaction time trials). We observed consistency in the reward rate dependence between the go cue and the movement onset period. So, we included both go cue and movement onset period as ‘action period’.

We further addressed this issue reconsidering our method to establish the latency of the movement onset. Initially, we defined the reaction time as the time when the lever left the central ‘holding’ area (20% of total lever movement range). We observed that substantial lever movement may occur before leaving the holding area. We corrected the reaction time definition as the point where the cumulative sum of the lever trajectory drifts more than three standard deviations beyond mean trajectory during the holding period (Fig. E2a; L-520). Using this approach, the activity change using the movement onset alignment was higher in most populations. Moreover, the reward rate dependence of the movement period activity was higher (or turned significant in those populations that were not before, i.e. DMS – dSPN and DLS – dSPN). We considered that the comments of the reviewer help us to improve the quality of the analysis. We really appreciate it.

To avoid confusion, we changed the term ‘action period’ and ‘action-related’ for ‘movement period’ and ‘movement-related’ in the main text (L-10, 13, 162, 217, 262, 278, 293, 303, 318, 356, 371, 377).

5. Would be useful to describe in Figures 4 and 5 what information is being provided in plots c and d in language that doesn’t require knowing what z score and r value are determined and what is different between what “c” and “d” provides in terms of information (clarify in the figure legend).

A. We made the proper clarifications in the figure legends and plot labels (L-1110, 1129).

6. dSPNs they report “higher activation in response to rewarded than non-rewarded outcomes”, what about during the “movement” onset period, (L-256) is the difference between the rewarded and non-rewarded considered to be “weak” for these examples (in comparison to the iSPN, ‘double peak’).

A. After revising our methodology for defining the movement onset (see our response to major comment 4), we replaced the go-cue alignment for the movement onset alignment. The dSPNs showed activation during movement onset (showing the ‘double peak’) in a similar fashion to iSPNs. We compared the peak activity of different identified projection neurons after Go cue and movement onset. Among the SPNs, the DMS-iSPNs exhibited the lower peak activity, being significantly lower than the DMS-dSPNs and DLS-dSPNs (Fig. E1e, f). In the case of DANs, there was no significant differences in the peak activities between rewarded or no-rewarded trials, or between DMS- and DLS-projecting DANs.

7. The statement that “only action- and outcome- related activity of DLS-dSPNs during non-rewarded trials were positively correlated to reward rate” needs to be explained in terms of what data they are using specifically to make this statement. – is this based on the histogram data in fig 4 “d” if so, need to provide more description (if we include summary figure in main figures, clarify the significance of each rew or no-rew).

A. After revising our methodology for defining the movement onset (see our response to major comment 4), the movement related activity exhibited higher activation, in addition to a stronger correlation to the reward rate, DMS-dSPN showed positive correlation with reward rate in both rewarded and no-rewarded trials. The DLS-dSPN movement- and outcome-related activity exhibited positive correlation with reward rate only in no-rewarded trials. We made the proper modification in Figure 4 and in the main text (L-277).

8. The results for the iSPNs need clarification: for DMS-iSPNs, in C) the response is significantly greater to the non-rewarded outcome tone than the rewarded outcome tone, but in D) looks like it is the opposite.

A. In Figure 5c we showed the average response to the outcome tone. Where the response to no-reward tone was higher than to the rewarded tone. In Figure 5d (right panel) both rewarded and no rewarded trials outcome-related activity showed positive correlation with reward rate. Additionally, the no rewarded trials showed a higher activity than the rewarded trials in each reward rate.

9. As exemplified by the above the difficulty of understanding what the data in Figures 4 and 5 are intended to demonstrate a better description of the experimental paradigm is required- exactly how is “reward experience” determined.

A. We include a clarification in the Results (L-140) and Methodology (L-682) section explaining the logic of choosing the reward rate as a measure of the reward experience in the past five trials.

Minor comments

In addition to comments above there are other problems with this study.

1. After multiple readings it was possible to figure out that the results of the “activity” in dSPNs and iSPNs during the task was presented in two ways, one was the actual “activity” of the SPNs and the other was a measure of the “activity” relative to the “reward expectation/reward rate”. This needs to be clarified, both in the set-up of the experiments, and in the description of the results.

A. Yes. We included the actual dependence of the activity on the reward rate, and also the degree of dependence of individual neurons using the correlation coefficient of the activity of each neuron with the reward rate. Additionally, we

compared the strength of the correlation between two or more populations, using the Fisher's z-test of the correlation coefficients with Bonferroni correction for multiple comparison. We clarified this issue in the Methods (L-684) section and in each figure legend.

2. The authors try to make the case that the since DA release in the striatum appears to be correlated with "reward expectation" that the value of the "activity measured relative to reward expectation/rate" is "dependent" on striatal DA release. While they might argue that they are correlated, this study does not provide evidence of "dependence".

A. We acknowledge that the study does not provide direct evidence of dependence between striatal DA release and activity relative to reward rate. However, we provide evidence for a correlation between these variables, suggesting that changes in reward rate may be reflected in the activity of striatal neurons that are modulated by DA release. The reward rate considering past events and the activity being current event. Also, while we do not show a causal relationship between striatal DA release and the activity measured, it is consistent with previous research showing that striatal DA release is involved in reward processing and that changes in reward expectation can modulate the activity of striatal neurons. Therefore, we consider reasonable that the correlation observed in the study between striatal activity and reward rate may be due, at least in part, to changes in striatal dopamine release. However, we also acknowledge that further studies would be necessary to establish a causal relationship between these variables.

3. The authors are aware of this as they state" The reward information processing reflected in the action-related activity may be under local striatal control and/or control from input other than dopamine." There are other possible sources not mentioned, including the thalamus and the globus pallidus in addition to the cortex, which is mentioned.

A. Certainly, we acknowledge the possibility that other brain regions beyond those mentioned in our text may be involved in modulating the activity of striatal neurons in response to changes in reward expectation. Previous works have shown that brain regions such as the anterior cingulate cortex, orbitofrontal cortex, thalamus, and globus pallidus, among others, are involved in regulating striatal activity in response to changes in reward expectation and other factors (Schilman et al., 2008; Jahfari et al., 2012; Matsumoto et al., 2015).

However, we wanted to focus on the activity of a specific set of neurons in the dorsomedial striatum. In future experiments, by analyzing the activity of these other areas in response to different behavioral parameters, we may understand better the specific role of these neurons in reward-based decision-making. We made the proper clarifications in the ‘Discussion’ section (L-339, 377).

4. The authors use the term “modulated” in 5 instances without providing evidence.

A. We made the proper change to ‘correlated’.

5. In the discussion there are several concerns with this sentence: “Indeed, the action-related activities of the identified DMS- and DLS-iSPNs were positively modulated by the reward experience, suggesting that reward experience-dependent action information is conserved throughout the mediolateral nigrostriatal system, mainly throughout the striato-pallidal pathway (Fig. 5).”

a. “modulated by” needs to be changed to “correlated with”

A. We made the proper corrections (L-356, 414).

b. Throughout the paper it is confusing “action related activities” are the same as “reward experience”, in this sentence they are used as though they are the same, but in other parts of the studies it seemed as though a distinction is made (such as in Figure 5C and 5D).

A. We clarified the terminology, changing the ‘action related activity’ for ‘movement related activity’. Also, we changed the expression ‘reward experience’ for reward rate, in order to avoid confusion.

c. In Figure 5C and 5D it appeared that there are differences for iSPNs between the DMS and DLS, but here it is suggested they are not different, clarification is needed.

A. We confirmed the dependency of neuronal activity of iSPNs in DMS and DLS. After performing multiple comparison of the correlation coefficients (degree of dependency on reward rate) among the four neuron populations, we observed significant difference between DMS and DLS, we made the proper statements in the results section and in Figure 5e.

6. In the Abstract and in the main text: clarification is needed of how “action-related activity”, “reward-expectation”, are defined and distinct “outcome-related” activity.

A. We include some clarification in abstract, changing ‘action-related activity’ for ‘movement-related activity’ (L-10, 13). Also, we specified what we mean with ‘outcome-related activity’ (L-10).

7. A major problem is the use of a “strawman” to contrast their findings with their view of a “classical model”. This is an overused construct in an attempt to claim experimental findings are novel. What specifically are the author’s referring to as the “classical model”. An example of the classic model of the basal ganglia is that the direct and indirect pathways originate from separate populations of SPNs, which respectively express the D1 and D2 dopamine receptors and that activity in the direct pathway promotes and activity in the indirect pathway suppresses movements. This “model” was originally proposed based on clinical disorders, such as Parkinson’s disease, where the absence of DA results in activity in the indirect pathway dominating to produce bradykinesia. Very soon after this model was proposed (and it might be more accurate to describe the

original as a model of clinical disorders) Mink (1996) proposed that normal motor behavior involves promotion of selected actions and suppression of competitive actions respectively by activity in the direct and indirect pathways. In keeping with this concept, numerous studies have shown that both dSPNs and iSPNs are active in a wide range of behavioral paradigms. As this conceptual framework is refined studies have expanded ideas of what constitutes selected and competing actions to recognize that activity in the direct and indirect pathways act both cooperatively and antagonistically.

A. While it is true that some studies might build a "strawman" version of the classical model to highlight the novelty of their findings, it's important to consider that the new models have built up based on this classical model, and some of the original ideas are still supported by recent studies. According to the classic theory of the basal ganglia, the direct and indirect pathways come from different populations of SPNs that express the D1 and D2 dopamine receptors, respectively. The activity in the direct pathway promotes movement while activity in the indirect pathway suppresses it. Numerous investigations in rodents and primates have supported this generally recognized model of the basal ganglia. For instance, some studies demonstrate that D1 and D2 receptor-expressing neurons have unique anatomical and physiological characteristics, and that manipulation of their activity can have differential effects on behavior (Kravitz et al., 2010; Freeze et al., 2013). Activating the direct pathway promotes movement while activating the indirect pathway suppresses it, and likewise, inhibiting the direct pathway inhibits movement while inhibiting the indirect pathway promotes it, according to research in rodents (Jin and Costa, 2010; Cui et al., 2013). Additionally, despite the fact that the classical model was used to explain the consequences of dopamine depletion in Parkinson's disease, it has since received widespread support from research in both healthy and pathological conditions. For example, studies in primates have shown that the direct and indirect pathways play a critical role in shaping movement-related activity in the motor cortex, and that manipulation of their activity can alter movement patterns (Nambu et al., 2002; Tachibana et al., 2011). In addition, research on humans have demonstrated that deep brain stimulation of

the subthalamic nucleus, a crucial component of the basal ganglia circuitry, can lessen Parkinson's disease's motor symptoms (Weintraub and Zaghoul, 2013).

In keeping with this notion, we believe that Mink's model extends the concept of the classical model without totally excluding it. It is true that the idea of the direct and indirect pathways as 'gas' and 'brake' is an oversimplification, as several studies have shown more complex interactions. For instance, the indirect route suppresses competing actions by inhibiting the thalamus, whereas the direct pathway supports certain actions by disabling thalamocortical circuits (Kravitz et al., 2010; Cui et al., 2013). The direct and indirect pathways, however, can work either cooperatively or antagonistically to shape behavior in a variety of contexts (Hikosaka et al., 2000; Jin and Costa, 2010). In the instance of our results, the fact that iSPNs are enhanced stronger than dSPNs by reward expectation may reflect the fact that the indirect pathway is involved in inhibiting unwanted movements, which may be necessary in a reward-based choice task where the animal needs to wait for a reward before initiating movement. This finding could be useful for the refinement of the classical model. Furthermore, our study shows that outcome-related activity was consistent with the classical model and reinforcement learning theory, which suggests that the classical model may be still relevant in explaining reward-based learning and decision-making under some contexts. Overall, while this study provides a new perspective on the role of the indirect pathway in reward-based choice tasks.

Responses to Reviewer #2's comments

This study analyzed the impact of dopaminergic signals in the activity of the direct and indirect striatal pathways in the context of forelimb movements and its rewarding consequences. Throughout a series of complex experiments combining a probabilistic task with fiber photometry and high-density electrophysiological recordings targeting the substantia nigra pars compacta (SNc) and the dorsolateral (DLS) and dorsomedial (DMS) striatum, the authors evaluated striatal dopamine release and the spiking activity of identified iSPNs, dSPNs, and SNc dopaminergic neurons during movement production and reward. The results may be divided in two streams. First, those related to movement initiation, where average dopaminergic activity (release, and SNc spiking activity), and most relevantly, striatal spiking activity in both, direct and indirect pathways, increased during movement production. These observations challenge the canonical views on function of the basal ganglia (BG) during movement production. And second, those results related to reward consequences, where dopaminergic neurons spiking activity in SNc and dopamine release in dorsal striatum increased after rewarded trials but decreased during non-rewarded ones. In my opinion, the experimental procedures and results are well designed and carefully performed. The observations may be of great relevance for our understanding of the BG, however, I still have some questions regarding the presentation and interpretation of these results.

Major comments

1. I believe that the title of the MS (general) does not entirely reflect the general findings of the work. It is completely biased to the observations related to reward expectation and indirect pathway. But in my opinion, another fundamental finding of this work is that, around movement initiation, there is a general dopaminergic-related cascade of events that includes an increase in dopaminergic release in the dorsal striatum and an increase in spiking activity in SNc spiking and both main striatal subpopulations iSPNs, dSPNs. While the specific consequences of this generalized activation will need to be thoroughly explored, the observations are at odds with what is expected at the onset of

movements from the classical model of the BG. In my opinion, these findings challenge validity of the general model of the BG during movement initiation.

A. We appreciate the suggestion. We modified the title: ‘Reward expectation enhances action-related activity of nigral dopaminergic and two striatal output pathways’.

2. Reward rate should be thoroughly explained in the results or methods section. I found that the general concept of Reward Rate is not entirely clear and required two or three times for the reader to finally grasp the idea that values closer to 0 means that the animals express a higher probability to switch behavior on the next trial, while values closer to 1 means that the animals tend to repeat the same behavior. I believe that clarifying this point is fundamental for the general understanding of the MS.

A. We made a clarification of how we quantified the reward experience using the reward rate (the mean of rewards obtained in the past five trials) (L-140). This clarification was made also in the Methods section (L- 682). Finally, we change the term 'reward experience' for 'reward rate' in the text if we judge it could generate confusion in the reader (L-88, 123, 125, 141, 152, 210, 220, 221, 223, 225, 231, 264, 281, 320, 323, 330, 332, 333, 364, 367, 677, 682, 683).

3. Is interesting that the activation latency of iSPNs and dSPNs are different at movement onset (Fig. 5 & 6C). Was this analyzed? What would be the reason for this difference? On the same line of thinking, dopaminergic spiking activity (Fig. 3) and dopaminergic release (Fig. 2) expressed a clear difference in latencies immediately after reward outcome (Fig. 2C & 3C). However, iSPNs/dSPNs latencies at reward outcome appear to be faster than dopaminergic release (Fig. 4C & 5C). What would be the reason for this apparent discrepancy? I believe that a systematic analysis of response latencies at movement onset and reward outcome for all experiments would help to clarify the dopaminergic dynamics on this behavioral context.

A. In order to compare the activation latency, first, we reconsidered our method to establish the latency of the movement onset. Initially, we defined the reaction time as the time when the lever left the central ‘holding’ area (20% of total lever movement range). We observed that substantial lever movement may occur before leaving the holding area. We corrected the reaction time definition as the point where the cumulative sum of the lever trajectory drifts more than three standard deviations beyond mean trajectory during the holding period (Fig. E2a). Using this approach, the activation during using the movement onset alignment was higher in most populations. Moreover, the reward rate dependence of the movement period activity was higher (or turned significant in those populations that were not before, i.e. DMS – dSPN and DLS – dSPN).

We compared the peak activity after the Go cue and movement onset of SPNs and SNc identified neurons. Overall, almost all population exhibited similar peak activity latencies after Go cue (Fig. E3a) and movement onset (Fig. E3b). The only instance that showed a significant shorter latency was the movement-related activity of DMS–SNc neurons (no-rewarded trials). Action-related activity in the dorsal striatum may be driven by robust inputs including the cortical and thalamic inputs (43, 53), and therefore, it might be less affected by the DA inputs. However, our findings showed a shorter latency of DMS-SNc neurons (Fig. S7), which could drive the movement-related activity of SPNs. Previous studies suggest that phasic dopamine release in the dorsal striatum is crucial for regulating movement intensity, where increased motivation is linked to increased actions to obtain a reward (54). Our study observed a similar relationship between movement-related activity in the DMS-SNc neurons and SPNs, which suggests a possible connection between dopamine levels and movement intensity under our experimental conditions. Also, the latency of the DANs firing activity was shorter than that of the dLight signal response, as expected considering the time needed for dopamine release and uptake (39).

4. In general, for the electrophysiological experiments, I found that the target populations ended up with significantly lower numbers than the total recorded neurons. For example, in SNc, authors reported more than 1500 neurons recorded (unidentified), but the dopaminergic neurons included in the final analysis were only 56 (33 and 23 projecting to DMS and DLS (identified), respectively; lines 167-168), that would be about 3.7 % of total recorded neurons. In the case of dorsal striatal SPNs, please provide specific numbers for each subpopulation, for the clustering clouds reported in SFig.5A, it appears that MSNs are very little. With that said, from the 2002 cells recorded in the striatum, the final numbers included in the analysis for dSPNs and iSPNs were 66 and 42, respectively. In both cases, striatum and SNC, the impact of the final small samples on the interpretation of the data must be discussed. (clarify for all populations).

A. We acknowledge that the number of identified neurons were significantly smaller compared to all the recorded neurons. There are several factors that explain this issue:

a. In order to be confident about the recorded SNc neurons identity, we used several criteria after performing the light-evoked collision test (64):

i. Constant latency of antidromically evoked spikes (jitter <1ms).

ii. Fixed frequency (frequency-following test, two pulses at 100–250 Hz).

iii. Collision test (44).

b. After the identification, we selected only neurons that were related to movement initiation.

c. Due to the fact that most neurons exhibited different activation during ‘pull’ or ‘push’ choice, we selected only neurons that had a spike rate difference of at least 5% between push and pull trials. We only considered the preferred action activity for the reward experience dependence analysis.

In order to confirm that the results of the identified neurons reliable represent the activity of all the recorded neurons we performed a similar analysis on putative neurons (selected using spike features and the coordinates of the location of the

recording probes, L-642). The dependence of the movement- and outcome related activity of the putative neurons were consistent with the optogenetically identified neurons (Fig. S3, S6).

We made the proper clarifications in the 'Methods' (L-585, 621).

5. In the same context, I found a bit surprising the lack of neurons decreasing their spiking activity during movement onset, especially in the striatum, where several previous publications suggest a higher response heterogeneity in this region, with neurons encoding kinematic, spatial and temporal parameters of specific movements. The fact that reaction times and velocity were nicely correlated with reward rate (Fig. 1F) can be used to further clarify the role of iSPNs, dSPNs and dopaminergic neurons. For example, what is the correlation of individual neurons in SNc and DLS/DMS with these parameters? Is it possible that dopamine and striatal dynamics are monitoring movement dynamics? I think it would be very useful to see the raster plots of a few representative neurons on each region. Include representatives.

A. Among all task related neurons there was a small percentage of neurons that decreased their activity after Go cue or movement onset (Fig. E4, 5 ,6). The criteria used for selecting movement-related neurons emphasized the activation around movement onset (and selecting the preferred activation between 'pull' and 'push'), this may exclude most of the neurons that decreased their activity around movement onset. We show the colormaps with all task related neurons (putative (Fig. E4, 5) and identified (Fig. E6)) with the proportion of neurons that were activated or suppressed around movement onset (Fig. E7).

We made some representatives raster plots visible to readers in Fig.S4. Additionally, we made proper clarifications in 'Results' (L-255).

B. We addressed the fact that RT and velocity were nicely correlated with the reward rate. First, we evaluated the correlation of individual neurons activity with

the reaction time and velocity. We found some individual neurons' Go cue-, movement-, and outcome-related activity that were positively or negatively correlated to reaction time. As suggested by the reviewer, we observed a substantial number of neurons whose Go cue- and movement-related activity correlated with reaction time, which may suggest that the activation during this period are monitoring movement dynamics. Among all the identified populations, the DLS-SNc Go cue-related activity and the DMS-SNc movement-related activity had the higher proportion of correlated units with reaction time (Fig. E8a). The distribution of r values indicated a higher number of units negatively correlated with reaction time, indicating the activation during movement is higher at shorter reaction times (Fig. E8b). However, during the outcome period, only a small number of units exhibited a correlation with reaction time or velocity (with not a clear bias to in the r values distribution[Fig. E8b)), suggesting that contrary to the movement-related activity, the activity during outcome period is not related to monitoring movement dynamics, but with monitoring the reward.

We made some clarification about this issue in 'Discussion' (L-355).

Minor Comments

1. Please indicate final numbers of IN and DAN. Gives the impression that DANs are significantly smaller population than IN (lines 153-156).

A. We recorded a total number of 1,523 midbrain: putative DANs ($n = 502$) and putative GABAergic interneurons ($n = 1021$). We then further selected the task related neurons (DANs: $n = 212$; INs: $n = 138$, see Methods). The putative DANs were further divided into medial ($n=107$) and lateral ($n=73$), based on the coordinates of the recording probes. This approach has some limitations: first, the coordinates alone are not a reliable way to identified a specific population; second, the medio-lateral segregation of DMS- and DLS- projecting SNc neurons is not clearly delimited; third, due to the anatomy of the SNc, we cannot discard that we

recorded surrounding SNr neurons (which exhibit a narrow spike waveform, so they were included in the putative GABAergic interneurons). Due to these limitations, we included this data as supplementary information, and we showed in the main results the reliably identified neurons using the light-induced collision test.

We made the proper clarifications about the neuron numbers in the main text (L-159, 163, 175, 181, 185, 251, 273).

2. How many neurons were classified as interneurons also produced antidromic spikes? If none, it would be important to state it (lines 167-168).

A. Indeed, some neurons classified as GABAergic interneurons exhibited some response to the light stimulation. However, this response did not comply with the criteria used for the analyzed neurons (i.e. fixed latency, small jitter, collision test). So, this response may indicate a synaptic response. We did not include this data in the analysis. We include this clarification in (L-187).

3. Please include representative raster plots of neural activity for each condition, this could be included in the main or supplementary figures.

A. We included the representative plots as Supplementary figure 4.

4. I found very useful the results summary displayed in supplementary figure 6, if consistent with the guidelines of the journal, I would suggest including it as last main figure.

A. We included the Supplementary figure 6 as main Figure 6.

Data not shown

Figure E1

Figure E2

Figure E3

Figure E4

Figure E5

Figure E6

Figure E7

Figure E8

a

b

c

Reviewers' comments:

Reviewer #1 (Remarks to the Author):

Review of

Reward expectation enhances action-related activity of nigral dopaminergic and two striatal output pathways

The authors' response to the "strawman" comment indicates that they appear to agree with the comment, and are aware of the many studies that have enhanced the original basal ganglia model. The point this reviewer was trying to make was that the many studies demonstrating concurrent activity in both the direct and indirect pathways during actions built on the original model of the neuroanatomical and functional differences between striatal neurons expressing D1 and D2 dopamine receptors, and that the activity in these pathways act both cooperatively and antagonistically.

In this study the authors are challenging a major component of the classic model, with their claim that dopamine acting through D2 receptors on iSPNs enhances and does not inhibit their activity. Considering the many (hundreds) of studies that have demonstrated opposite effects of dopamine on dSPNs and iSPNs, theirs would be an important finding if supported by data. Unfortunately, the data as presented are not convincing to support such a finding.

Their findings are based on the following logic:

1) A task is used in which after a "go cue" a rat chooses to either push or pull a lever (movement), 300 msec after the go cue, either a high or low tone is provided (a high tone indicating a water reward will be given 70% of the time and a low tone indicating no-reward with water provided only 10% of the time). The "reward tone" or "no-reward tone" is linked to either the "push" or "pull" movement in different blocks of trials. After training the rats typically execute the "correct" push or pull movement to obtain a reward. The association of the "push/pull" movement with the "reward/no-reward" outcome was switched when an animal selected the "high option reward" movement 79% of the time in the previous 10 trials. On average this switch occurred after 30-50 trials.

2) This "probabilistic" push/pull task was used as the animals "choice" was influenced by the mean number of rewards obtained in the prior 5 trials. On average if an animal received no rewards on 2 trials in a row they would switch to the other push/pull option. This relationship between the number of "reward" trials in the prior 5 trials was defined as the "reward rate".

Question: in the probabilistic version, in the "high value" block of trials, when rewards are given 70% of the time, is the "high tone-reward" tone given on trials when there is no reward?

Question: "reward rate" is the most important measure used in this study and it is not clear how it is calculated: all that is stated is that it is the "average number of rewards over the last five trials". But in the figures it is provided in a range between 0 and 1. This does not make sense, how are the values between 0 and 1 calculated (at the least it seems as though the value range must differ between "rewarded" and "non-rewarded" trials.

3) During the task measures were taken of DA release in the striatum, activity of DA neurons in the SNc and activity of dSPNs and iSPNs. Measures were determined throughout the task but tied to 3 specific events: Go cue, initiation of movement and outcome tone presentation.

Question: Since the "Go cue" initiates the animal's movement, the two are typically within a similar period and wonder why they aren't shown together. Is it that there is a difference in the delay after the go-cue and when the animal initiates movement? Seems as though this was stated but not clearly.

4) During the task time-course, DA release in the striatum increases at the movement onset for approximately 300 msec (during the period when the rat is holding the lever in place after moving)

Question: In Figure 2 C why are the plots different for the "Go Cue" and "movement onset", shouldn't they be the same except for the movement onset being set to the time when movement is initiated? Similarly does the plot of the DA signal in the "Outcome tone" plot during the -0.5 to 0.0 time period the same as what is plotted in the Movement onset plot in the 0.0-0.5 time period. This is confusing,

and should be better explained.

Question: In Figure 2 C: Need to explain more about what the plots of "r-values" to Reward Rates show, what is the meaning of differences in the slope of the lines for positive (movement onset) and negative (Outcome tone)? More explanation of how what is being claimed for boxplots also needed.

Question: The increase in DA release in the striatum starts to increase (in the reward trials) somewhat delayed to the onset of the outcome tone (250 msec), discussion is needed as to whether this reflects the time delay between the activity of the SNc DA neurons or some other factor(s).

Question: In Figure 2 C in the r value plots for the "Go Cue" and Movement Onset" periods it does not appear that there is much of difference between the "rewarded and non-rewarded" trials, although the authors seem to be claiming a "dependence on DA release and reward rate". More explanation is needed for what these plots are showing and how they support the conclusions stated.

5) They compute a z-score value of the relationship between reward rate and activity of dSPNs and iSPNs during different periods of the task. They assert that their data demonstrate that contrary to the classic model, dopamine increases activity of iSPNs. There are several problems with this interpretation of the data.

a. How "reward rate" is computed is unclear and as this is the critical value that their conclusions are based on this needs to be defined. Reward rate is defined as the average number of rewards in the prior 5 trials. What is not clear is how a range between 0 and 1 could be obtained. For example how is a reward rate of 1.0 obtained for non-rewarded trials, when the number of rewarded trials is less 10% or less. Without clarification it is not possible to understand how "reward rate" is related to any of the measures.

b. The increase in iSPN activity tied to the initiation of movement (in fig 5c) precedes the increase in DA release in the striatum (in Fig 2c middle) so that it is hard to make the case that iSPN activity is "dependent" on DA.

c. For the "outcome tone" period the highest level of activity is in the 100 msec following tone delivery after which the activity in iSPNs decreases (Fig 5c). In this time period striatal DA release does not start to increase for 150 msec or so into the time period, at which time the activity of iSPNs is suppressed. This does not support the idea that DA enhances iSPN activity.

d. The "reward rates" correlations to activity levels are calculated for the 500 msec time from the beginning of the "outcome tone", during this time the activity levels go through various phases, first increasing briefly and then decreasing. An average number obtained would not necessarily reflect the effect of DA.

e. For the "go cue", "movement initiation" and "outcome tone" period action related activity is reported to be positively modulated by dopamine in both the "rewarded" and "nonrewarded" trials, which do not appear to be directly correlated with DA release in the striatum. This does not support the idea that such activity is dependent on DA.

6) There are a number of conceptual issues.

a. In their response to reviewer comments that authors acknowledge that their data does not provide evidence of causal relationships. Despite this they continue to use "dependence" in all the figures and text (44 times) when at most the data provide a correlation and in many cases a lack of correlation between different measures.

b. It is well established that DA by itself is not responsible for SPN activity but rather modulates the response to cortical and thalamic glutamatergic inputs, which does provide the input that produces action potentials (activity) in SPNs. The most likely driver of activity in both types of SPNs in the task studied is glutamatergic inputs from cortex or thalamus. Since that activity in both dSPNs and iSPNs following the "go cue" and "movement initiation" precedes increases in striatal DA, this activity appears to be not dependent on DA.

c. The authors do refer to corticostriatal inputs in manner that suggests that such input may only have a minor affect (referencing "pyramidal tract-type neurons in the motor cortex emit collaterally to the dorsal striatum - 46) while PT inputs to SPNs are considerably less abundant than intratelencephalic (IT) corticostriatal projections, which are likely to be responsible for the activity being recorded during the task.

7) And to repeat that this study does not provide evidence of the dependence of any of their measures on "reward rate".

Reviewer #2 (Remarks to the Author):

The authors thoroughly addressed and clarify all my concerns and recommendations. I'm very satisfied with the state of the MS, I found the new version very round and solid and suitable for publication.

Review of:

Reward expectation enhances action-related activity of nigral dopaminergic and two striatal output pathways

The authors' response to the "strawman" comment indicates that they appear to agree with the comment and are aware of the many studies that have enhanced the original basal ganglia model. The point this reviewer was trying to make was that the many studies demonstrating concurrent activity in both the direct and indirect pathways during actions built on the original model of the neuroanatomical and functional differences between striatal neurons expressing D1 and D2 dopamine receptors, and that the activity in these pathways act both cooperatively and antagonistically.

In this study the authors are challenging a major component of the classic model, with their claim that dopamine acting through D2 receptors on iSPNs enhances and does not inhibit their activity. Considering the many (hundreds) of studies that have demonstrated opponent effects of dopamine on dSPNs and iSPNs, theirs would be an important finding if supported by data. Unfortunately, the data as presented are not convincing to support such a finding.

The original and previous revised versions of our manuscript were not clearly written and actually in a way that could easily be misinterpreted in some parts. We sincerely apologize for this and appreciate the reviewer's valuable and helpful comments.

In our study, we observed that "reward expectation" level based on recently obtained rewards (quantified as reward rate) enhanced the iSPN activity. However, we have never argued "dopamine (DA)" enhances iSPN activity (via D2 receptors). Assuming the classical basal ganglia model, we concluded that the reward expectation acts on the iSPN activity probably through an effect of cortical or other input rather than a direct effect of DA. This was also indicated by the thickness of light green arrows in Figure 6 schema. We believe that it was exactly consistent with the reviewer's suggestion.

In this revision, we have made appropriate improvements to our manuscript (for example, please see corrections in the Abstract) in accordance with each of the reviewer's comments.

Their findings are based on the following logic:

- 1) A task is used in which after a "go cue" a rat chooses to either push or pull a lever (movement), 300 msec after the go cue, either a high or low tone is provided (a high tone indicating a water reward will be given 70% of the time and a low tone indicating no-reward with water provided only 10% of the time). The "reward tone" or "no-reward tone" is linked to either the "push" or "pull" movement in different blocks of trials. After training the rats typically execute the "correct" push or pull movement to obtain a reward. The association of the "push/pull" movement with the "reward/no-reward" outcome was switched when an animal

selected the “high option reward” movement 79% of the time in the previous 10 trials. On average this switch occurred after 30-50 trials.

We thank the reviewer for the review and question. In our task design, high and low tone cues indeed correspond directly to the outcome of the trial, reward and no reward, respectively. For example, in the 70% push rewarded block (as illustrated in Fig. 1a), if the rat pushed the lever, a high-tone was given at 70% probability, and the high-tone (10 kHz, 60 dB, 0.3 s) always brought a reward. But in case of a low-tone (4 kHz, 60 dB, 0.3 s), which was given at remaining 30% for push, the low-tone always led to no reward. In contrast, if the rat pulled the lever, the high-tone was given at 10%, and the high-tone always brought reward. But in case of the low-tone, which was given at 90% for pull, the low-tone always led to no reward.

Thus, the high tone always signaled a reward, and the low tone always signaled a lack of reward.

We realize now that this was not clearly explained in our original manuscript, although Fig. 1a was a correct diagram. We appreciate the reviewer bringing it to our attention. We have revised the relevant sections of Results (L.96-102), Methods (L.555-561), and Fig. 1a legend (L.1014-1017) to clarify this point.

2) This “probabilistic” push/pull task was used as the animal’s “choice” was influenced by the mean number of rewards obtained in the prior 5 trials. On average if an animal received no rewards on 2 trials in a row they would switch to the other push/pull option. This relationship between the number of “reward” trials in the prior 5 trials was defined as the “reward rate”.

Question: in the probabilistic version, in the “high value” block of trials, when rewards are given 70% of the time, is the “high tone-reward” tone given on trials when there is no reward?

As we mentioned in the above response, in our task design, the high and low tone cues correspond directly to the outcome of the trial, reward and no reward, respectively. The reward (water) was delivered with a delay of 0.3 s, Methods (L.555-561).

Thus, again, the high tone always signaled a reward, and the low tone always signaled a lack of reward. We have revised the manuscript as described above.

Question: “reward rate” is the most important measure used in this study and it is not clear how it is calculated: all that is stated is that it is the “average number of rewards over the last five trials”. But in the figures, it is provided in a range between 0 and 1. This does not make sense, how are the values between 0 and 1 calculated (at the least it seems as though the value range must differ between “rewarded” and “non-rewarded” trials.

We apologize for any confusion regarding the calculation of the "reward rate." The reward rate is indeed calculated as the average number of rewarded outcomes over the last five trials (not including the current trial, which is used for dividing the plots into ‘rewarded’ and ‘non-

rewarded'). It is expressed as a proportion by dividing the number of rewards by five, which explains why the value is presented in the range between 0 and 1.

To clarify, the calculation works as follows:

For each trial, we look at the previous five trials and count how many were rewarded. This number (between 0 and 5) is then divided by the total number of the previous trials we're considering, in this case, five. This division essentially converts the count of rewarded trials to a proportion.

Hence, if all five of the last five trials were rewarded, the reward rate would be $5/5 = 1$. If none were rewarded, the reward rate would be $0/5 = 0$. And, if, for example, three of the last five trials were rewarded, the reward rate would be $3/5 = 0.6$.

We used this measure to capture the recent reward history of the animal, providing a relative scale to evaluate how rewarding it has been. The range between 0 (no rewards) and 1 (all trials rewarded) allows for a normalized comparison across animals and sessions.

We made sure to clarify this calculation in the Results (L.117-125; 131-133) and Methods (L.723-728), so it's easier for the reader to understand.

3) During the task measures were taken of DA release in the striatum, activity of DA neurons in the SNc and activity of dSPNs and iSPNs. Measures were determined throughout the task but tied to 3 specific events: Go cue, initiation of movement and outcome tone presentation.

Question: Since the "Go cue" initiates the animal's movement, the two are typically within a similar period and wonder why they aren't shown together. Is it that there is a difference in the delay after the go-cue and when the animal initiates movement? Seems as though this was stated but not clearly.

In our task design, we distinguish three key events: Go-cue, the initiation of movement, and the outcome tone presentation. Even though the Go-cue signaled the rat to initiate movement, there was a variable delay before the movement actually began corresponding to the animal's reaction time. Thus, while the Go-cue and movement onset were related, they neither occurred simultaneously nor with a constant delay, and we therefore analyzed them as separate events.

To evaluate the possible variability of latency of movement onset and its potential effect on neural activity, we analyzed the difference in activation latencies at the Go-cue onset and movement onset for all experiments. We consider these results were not remarkable enough to include them in the manuscript, but we would like to highlight those findings here. When comparing the peak activity after the Go cue and movement onset among identified SPNs and SNc neurons, most populations exhibited similar peak activity latencies after Go cue and movement onset (Fig. S7). The only instance that showed a significantly shorter latency was the movement-related activity of DMS-SNc neurons in no-reward trials (Fig. S7), having a

potential role facilitating action initiation or modulating the vigor of actions. Also, it's worth noting that the latency of the DANs firing activity was shorter than that of the dLight signal response (compare Fig. 2c with Fig. 3c), which is consistent with the time required for dopamine release and uptake.

This explanation should help to clarify that there is indeed a delay between the "go cue" and the initiation of movement and provide some context about the differences in activation latencies between different populations of neurons. Also, the population activity of each neuron type, can be seen in the supplementary data (Fig. S2).

4) During the task time-course, DA release in the striatum increases at the movement onset for approximately 300 msec (during the period when the rat is holding the lever in place after moving):

Question: In Figure 2 C why are the plots different for the "Go Cue" and "movement onset", shouldn't they be the same except for the movement onset being set to the time when movement is initiated? Similarly does the plot of the DA signal in the "Outcome tone" plot during the -0.5 to 0.0 time period the same as what is plotted in the Movement onset plot in the 0.0-0.5 time period. This is confusing and should be better explained.

The differences between the "Go Cue" and "Movement onset" plots in Figure 2C can be attributed to the variable delay between the Go cue and the actual initiation of movement due to the rat's reaction time. The "Go cue" signals the rat to initiate movement, but the actual movement onset happens after a short and variable delay. This results in differences between the two plots.

As for the second part of the question, the "Outcome tone" plot indeed includes the same time period as the "Movement onset" plot; the data are overlapped but aligned differently. In the "Outcome tone" plot, the time 0 corresponds to the onset of the outcome tone, which means that the -0.5 to 0.0 sec period corresponds to the time just before the outcome tone, which includes the movement onset and the subsequent time when the rat is holding the lever in place. In contrast, in the "Movement onset" plot, time 0 corresponds to the movement onset, so the 0.0-0.5 sec period corresponds to the time just after the movement onset, which includes the same time period as in the "Outcome tone" plot but aligned differently. In short, it is actually biphasic.

We agree that this can be confusing, and we've clarified these points in the Results (L.138-140; 143-150) and Fig. 2b legend (L.1,053-1,058).

Question: In Figure 2 C: Need to explain more about what the plots of "r-values" to Reward Rates show, what is the meaning of differences in the slope of the lines for positive (movement onset) and negative (Outcome tone)? More explanation of how what is being claimed for boxplots also needed.

Thank you for drawing my attention to this issue as it is crucial for the clear communication and understanding of the research findings. We implemented a specific method for determining the correlation of DA release (Fig. 2)/neuronal activity (Figs. 3-5) with reward rate. The reward rate was calculated as the mean reward number of the previous five trials, i.e., the number of rewards divided by five (assigning a value of 1 to rewarded trials and 0 to non-rewarded trials), irrespective of the lever selection. This allowed us to quantify the average rewards received recently, providing a window into the animal's immediate reward history.

To assess the relationship between this reward rate and DA/neuronal activity (expressed as a z-score), we used Pearson's correlation coefficient, a standard measure of linear correlation between two datasets. We plotted these correlations, with the reward rate on the x-axis and the corresponding z-scored DA/neuronal activity on the y-axis. The slope of the lines in these plots represents the strength and direction of the correlation; positive slopes indicate that DA/neuronal activity increases with higher reward rate (as seen in the Movement onset plot), while negative slopes suggest the activity decreases with higher reward rate (as seen in the Outcome tone plot).

Additionally, we evaluated the degree of correlation with reward rate for individual sessions (Fig. 2) or neurons (Figs. 3-5). For this, we calculated the correlation coefficient (r) between the DA/neuronal activity and reward rate for each session/neuron.

In the boxplots shown in Fig. 2C, each box represents the distribution of correlation coefficients for individual DA measurements at a specific task event. The box extends from the lower to higher quartile values of the data, with a line at the median, providing a visual representation of the interquartile range and data dispersion. The 'whiskers' extend from the box to show the range of the data. Outlier points are those that fall outside the whiskers. Figures 3-5 are also the same for particular neuronal populations.

The boxplots illustrate a trend in the distribution of correlation coefficients, providing insight into the population-level relationship between reward rate and DA/neuronal activity. A higher or lower median bias indicates a stronger positive or negative correlation with the reward rate for the respective neuronal population. This method allows for a comprehensive overview of the differential effects of reward rate on the activity of the different neuronal populations during distinct task events as summarized in Fig. 6.

We have made the proper clarification in the Methods section (L.728-743).

Question: The increase in DA release in the striatum starts to increase (in the reward trials) somewhat delayed to the onset of the outcome tone (250 msec), discussion is needed as to whether this reflects the time delay between the activity of the SNc DA neurons or some other factor(s).

Indeed, an important observation in our results is the noticeable delay in the increase of DA release in the striatum following the outcome tone in reward trials (around 250 msec). One could attribute this delay to a few contributing factors.

Firstly, it's important to consider the temporal dynamics of DA neurotransmission. DA neurons in the SNc fire action potentials that lead to the release of dopamine in the striatum. There is a well-established delay between the initiation of action potentials in DA neurons and the subsequent increase in extracellular DA concentration in the striatum, which is due to the time required for vesicular release, diffusion, and detection by dLight 1.3b. This process can result in a latency in the observed DA signal (Chergui, 1994 (39); Mohebi, 2019 (37)).

Secondly, apart from the direct pathway from SNc to the striatum, the striatal activity and consequently, DA release, can also be modulated by inputs from other areas such as the cortex and the thalamus. The processing times in these brain areas, the propagation of the signals, and the interaction of these signals at the level of the striatum could contribute to the observed delay.

Further investigations would be valuable to elucidate the precise roles and timing of these various contributing factors. While our data provides a valuable insight into the dynamics of DA release during task performance, it also raises compelling questions that warrant future studies.

We have mentioned this issue in the Results (L.209-215; 307-312) and Discussion (L.353-364; 434-455)

Question: In Figure 2 C in the r value plots for the “Go Cue” and Movement Onset” periods it does not appear that there is much of difference between the “rewarded and non-rewarded” trials, although the authors seem to be claiming a “dependence on DA release and reward rate”. More explanation is needed for what these plots are showing and how they support the conclusions stated.

We apologize if our explanation was not clear enough and led to some misunderstanding. We'd like to clarify that the 'reward rate' used in our study is computed as the mean number of rewards obtained in the last five trials. This computation does not include the outcome of the current trial; hence the reward and no-reward trials are categorized based on the outcome of the current trial, independent of the reward rate.

The r-value plots for the 'Go Cue' and 'Movement Onset' periods in Figure 2c show the correlation between the reward rate (based on the past five trials) and the currently observed dopamine (DA) release. The reviewer is correct in observing that there is little difference between rewarded and non-rewarded trials during these periods. This is because DA release at these time points is more tied to the reward expectation based on recent rewards (represented by the reward rate) than the actual near-future outcome (reward or no reward; but still unknown) of the current trial.

Our claim for a 'dependence of DA release on reward rate' (not on current outcome) was based on these observations, showing that the level of DA release during these periods (go-cue and movement onset window) is influenced by the animal's recent reward history, and not by the subsequent outcome of the current trial. We have specified that the division was performed depending on current trial outcome in Results (L. 146), Figure 1 (L. 1066). We hope this explanation clarifies your concerns.

5) They compute a z-score value of the relationship between reward rate and activity of dSPNs and iSPNs during different periods of the task. They assert that their data demonstrate that contrary to the classic model, dopamine increases activity of iSPNs. There are several problems with this interpretation of the data.

As we mentioned above, first of all, we do not claim 'dopamine increases activity of iSPNs'. Rather, we believe that the correlation between recent rewards and current neuronal activity cannot be explained by dopamine alone (as illustrated in Fig. 6). We apologize for any confusion or lack of clarity in our manuscript. We have addressed the following issues point by point.

- a. How “reward rate” is computed is unclear and as this is the critical value that their conclusions are based on this needs to be defined. Reward rate is defined as the average number of rewards in the prior 5 trials. What is not clear is how a range between 0 and 1 could be obtained. For example, how is a reward rate of 1.0 obtained for non-rewarded trials, when the number of rewarded trials is less 10% or less. Without clarification it is not possible to understand how “reward rate” is related to any of the measures.

In our study, 'reward rate' refers to the mean number of rewards obtained in the past five trials, and it is an important measure to understand the anticipation of the reward by the animal. This does not include the outcome of the current trial, so the reward and no-reward trials are separated based on the outcome of the current trial, independent of the reward rate.

The computation results in a reward rate that is a decimal value between 0 and 1, with 0 indicating that no rewards were obtained in the last five trials and 1 indicating that rewards were obtained in all of the last five trials. This allows us to understand how the previous reward history affects the animal's current behavior and neurobiological responses. It is possible for a no-reward trial (current) to have a high reward rate if the animal received rewards in the previous trials, and vice versa.

The question about obtaining a reward rate of 1.0 in non-rewarded trials, when the number of rewarded trials is less than 10%, seems to stem from a misunderstanding. The reward rate for a non-rewarded trial can indeed be 1.0 if all five of the preceding trials were rewarded. This would indicate that the animal had a high recent success rate but did not receive a reward in the current trial.

We apologize if this was not clearly communicated in our manuscript and appreciate your attention to detail. We ensured to explain this more thoroughly in our manuscript revisions, the Results (L.117-125; 131-133) and Methods (L.723-728).

- b. The increase in iSPN activity tied to the initiation of movement (in fig 5c) precedes the increase in DA release in the striatum (Fig 2c middle) so that it is hard to make the case that iSPN activity is “dependent” on DA.

We do not consider that 'iSPN activity is dependent on DA', but rather 'on recent rewards', and iSPN may be modulated by other inputs; i.e. cortex thalamus, etc. (please see the above responses, too). We apologize if our statements were not clear enough in the manuscript.

We agree with the reviewer's important point that iSPN activity preceded DA release. This is exactly the evidence that iSPN activity is not driven solely by DA. We come to consider that inputs from other regions, such as the cortex or thalamus, might play significant roles in modulating the activities of both striatum and dopaminergic neurons in the midbrain during this time frame (see Fig. 6).

We have made this point clear in the Results (L.309-312).

- c. For the “outcome tone” period the highest level of activity is in the 100 msec following tone delivery after which the activity in iSPNs decreases (Fig 5c). In this time period striatal DA release does not start to increase for 150 msec or so into the time period, at which time the activity of iSPNs is suppressed. This does not support the idea that DA enhances iSPN activity.

The reviewer is correct in stating that the onset of iSPN activity reduction appears to coincide with the time point when striatal DA release begins to increase following the outcome tone, Results (L.306). The temporal dynamics indeed presents a nuanced scenario, which we did not explicitly address in our initial manuscript.

However, it is important to clarify that our claim is not that dopamine directly causes an instantaneous decrease in iSPN activity. Rather, we observed a positive correlation between dopamine release and iSPN activity in relation to the reward information over a broader time window (recent 5 trials), not on a moment-to-moment basis; i.e. only last or current trial.

Additionally, it is worth mentioning that dopamine effects are multifaceted and context-dependent. Dopamine can have both fast and slow effects, impacting different aspects of neuronal function, including excitability, synaptic plasticity, and long-term changes in gene expression.

We understand that our manuscript can be improved by providing a more nuanced discussion on these aspects, and we have revised the Discussion (L.353-364) to more clearly communicate these points.

- d. The “reward rates” correlations to activity levels are calculated for the 500 msec time from the beginning of the “outcome tone”, during this time the activity levels go through various phases, first increasing briefly and then decreasing. An average number obtained would not necessarily reflect the effect of DA.

We understand this point about the changing nature of neuronal activity during the 500 msec window following the onset of the "outcome tone".

However, our analysis was designed to evaluate the overall response pattern within this period rather than moment-to-moment changes. We acknowledge that the instantaneous activity levels do vary during this period, and thus, the averaged activity may not precisely reflect the effect of dopamine at every single point within this time window. However, it serves as an approximate measure to compare the overall response magnitude across different trials and conditions, allowing us to observe general trends and correlations with reward rate.

- e. For the “go cue”, “movement initiation” and “outcome tone” period action related activity is reported to be positively modulated by dopamine in both the “rewarded” and “nonrewarded” trials, which do not appear to be directly correlated with DA release in the striatum. This does not support the idea that such activity is dependent on DA.

We would like to clarify that our claim is not that iSPN activity is directly dependent on DA release. We agree with the reviewer’s pointing out that the correlations we see between dopamine levels and iSPN activity do not necessarily mean that one directly causes the other. The relationship we found might be indicative of an indirect or modulatory role of dopamine, or they might both be influenced by a common third factor.

In fact, as the reviewer has mentioned, other inputs, such as from the cortex or thalamus, may also be important for modulating the activity of iSPNs during the periods (see Fig. 6). We should have made this point clearer in our initial manuscript, and we have revised it accordingly to more precisely describe the nature of the relationship we observed between dopamine levels and iSPN activity, Results (L.366-382).

- 6) There are a number of conceptual issues.
 - a. In their response to reviewer comments that authors acknowledge that their data does not provide evidence of causal relationships. Despite this they continue to use

“dependence” in all the figures and text (44 times) when at most the data provide a correlation and in many cases a lack of correlation between different measures.

We appreciate the reviewer’s suggestion about the usage of the term “dependence”, and fully understand the concern that this term may imply causality, which as the reviewer correctly noted, our study does not directly provide.

The term “dependence” was used in our manuscript in a statistical sense, to indicate that one variable changes in relation to another. It was not intended to suggest a causal relationship. However, we understand how this could be misinterpreted and we agree that clarity is paramount.

To avoid any misunderstanding, we have revised the throughout manuscript to use terms such as “correlation”, “relationship” or “association” in place of “dependence” where appropriate (including most of headlines and figure captions). Now we can see the term “dependence” in only a few times, while “correlation/correlated” more than 70 times. We believe this will more accurately reflect the nature of the relationships we found in our data, without implying causality; Results (L. 129, 131, 135, 156, 159, 163, 168, 226, 228, 230, 231, 234, 237, 240, 248, 276, 281, 295), Discussion (L. 341, 350, 357, 395, 398, 408); Figure legends (L. 1051, 1096, 1136, 1149, 1167, 1239, 1320).

- b. It is well established that DA by itself is not responsible for SPN activity but rather modulates the response to cortical and thalamic glutamatergic inputs, which does provide the input that produces action potentials (activity) in SPNs. The most likely driver of activity in both types of SPNs in the task studied is glutamatergic inputs from cortex or thalamus. Since that activity in both dSPNs and iSPNs following the “go cue” and “movement initiation” precedes increases in striatal DA, this activity appears to be not dependent on DA.

We acknowledge that the reviewer's assertion aligns with established knowledge that dopamine (DA) is not the sole driver of SPN activity but rather modulates the response to cortical and thalamic glutamatergic inputs, which primarily elicit action potentials in SPNs. We absolutely agree with the view that glutamatergic inputs from the cortex or thalamus are likely the main instigators of activity in both dSPNs and iSPNs during the task studied.

In our paper, we have not made the claim that the activity of dSPN and iSPN is entirely dependent on DA. What we strive to handle and emphasize in our study is the direction and degree of its possible modulation, not its direct responsibility for SPN activities. In fact, the activity in both dSPNs and iSPNs following the “go cue” and “movement initiation”, which indeed preceded increases in striatal DA, appeared to be not directly dependent on DA. We have added this observation in the Results (L.307-312), which is in line with the reviewer's argument.

As we explain in our discussion (L.355-364), our data indicate that DA's modulation of iSPN activity does not occur instantaneously but rather in correlation with the patterns of dopamine release over a broader time window (recent five trials). This temporal relationship is complex and may not align perfectly on a moment-to-moment basis, further supporting the idea of multifaceted dopamine influences on neuronal function, which are context-dependent. Anyway, there is no evidence that the reward expectation is mediated through this slow DA effect over a broader time window. Therefore, while we observed a correlation between dopamine release and iSPN activity, we concur with the reviewer that the precise temporal relationship remains to be fully elucidated.

As illustrated in Fig. 6 and mentioned in the Discussion (L.366-382), the processing of reward information, as reflected in action-related activity, may indeed be under the control of local striatal mechanisms and/or inputs from regions other than dopamine, such as the anterior cingulate cortex, orbitofrontal cortex, motor cortex, thalamus, globus pallidus, etc. This viewpoint, as the reviewer rightly points out, further underscores the complexity and multifactorial nature of these neuronal activities.

In conclusion, we absolutely agree with the reviewer's point, and we thank them for providing the opportunity to clarify our position and to emphasize that our findings are intended to contribute to the existing knowledge of this complex interplay, rather than challenge it. We believe that a nuanced understanding of the role of dopamine in modulating SPN activity can advance our knowledge in this field, and our observations are offered in this line. Therefore, we discuss this issue further in our manuscript, Discussion (L.355-382). We have also revised the Abstract while keeping to the word limit (L.8-21).

- c. The authors do refer to corticostriatal inputs in manner that suggests that such input may only have a minor affect (referencing “pyramidal tract-type neurons in the motor cortex emit collaterally to the dorsal striatum – 46) while PT inputs to SPNs are considerably less abundant than intratelencephalic (IT) corticostriatal projections, which are likely to be responsible for the activity being recorded during the task.

It was not our intention to suggest that these corticostriatal inputs might have a minor effect. Rather, we referenced the contribution of pyramidal tract-type neurons in the motor cortex that emit collaterally to the dorsal striatum, Results (L.366-370) as one example among several potential sources of input that could modulate SPN activity.

We concur with the reviewer's assertion regarding the considerable impact of intratelencephalic (IT) corticostriatal projections on SPN activity. In fact, we agree that these projections are likely to be responsible for much of the activity being

recorded during the task. It is well-documented that these IT corticostriatal projections constitute a large portion of inputs to SPNs and are pivotal for information processing in the striatum.

In no way did we intend to diminish the significance of IT corticostriatal projections. We apologize if our original phrasing suggested otherwise. We modified our discussion to clearly articulate the prominent role of these inputs (L.370-377). We have revised our work to accurately reflect the importance of IT corticostriatal inputs in driving SPN activity. This feedback contributes substantially to improving the clarity and accuracy of our study.

7) And to repeat that this study does not provide evidence of the dependence of any of their measures on "reward rate".

Finally, we agree that our investigation does not establish a direct and absolute dependence of our measures on "reward rate". It does, however, provide strong indications of a relationship, based on correlations and patterns observed across our experiments.

The aim of our study was not to establish a direct causative relationship but rather to highlight correlations and potential associations that can help inform and advance our understanding of the neurobiology of reward processing. We believe that the patterns and correlations we observed contribute valuable insights to the field, and they are important stepping stones for future investigations. Our results shed light on the complex dynamics of neuronal activity in relation to reward rate and highlight potential areas for more targeted exploration.

We agree with the reviewer that further experimental and analytical work is needed to provide more definitive evidence for a causal relationship between our measures and reward rate. Such studies could involve targeted manipulation of reward rate and closer examination of the effects on neuronal activity, among other possibilities. Our study serves as a strong foundation for such future research.

We recognize the complexity of the reward system and that multiple factors and pathways likely interact to drive the observations we have made. Therefore, we do not claim that "reward rate" is the only or the primary determinant of the neuronal activities measured. Instead, we present it as a crucial factor that contributes to our understanding of the complex neurobiological responses to reward stimuli.

We hope this clarification addresses the reviewer's concerns, and we are grateful for the opportunity to enhance the description of our study's scope and its contributions to the field.

REVIEWERS' COMMENTS:

Reviewer #1 (Remarks to the Author):

see attachment

The manuscript is much improved. The experimental design is considerably easier to understand, and the description of various measures much clearer. A major criticism was that in the prior versions, the authors suggested that dopamine increased activity of D2 expressing iSPNs without evidence. The authors are now stating that "We do not consider that 'iSPN activity is dependent on DA', but rather 'on recent rewards', and iSPN may be modulated by other inputs; i.e. cortex thalamus, etc.". However, as written are likely to draw the conclusion that dopamine increases iSPN activity mainly because that is the way the study is structured.

The study first finds that "dopamine" activity and release is correlated with "reward rate" and then that "reward rate" is correlated with iSPN activity.

The data in Figure 5c shows that iSPN activity increases prior to DA release, which is what is stated, but then it is unclear what is meant by "suggesting possible common inputs other than dopamine". It actually suggests that iSPN activity might not be driven by dopamine at all. The text reads as though dopamine is the primary player in the patterns of activity, while the data suggest it might be a secondary player.

The most confusing part of this study is the data shown in Figure 5. As stated iSPN activity in rewarded trials increases prior to DA release, and then decreases. This suggests and is stated that after reward outcomes iSPN activity is suppressed. However, then in 5d and e, measures are shown that there is a correlation of iSPN activity with reward rate. The authors explain that while there the relationship between iSPN activity and the time course of the behavior "presents a nuanced scenario" that is due to the measure being averaged over a longer time period (that encompasses both the very transient increase in iSPN activity that appears unrelated to dopamine and the longer period of suppression that occurs after dopamine release begins). One might interpret the finding that the iSPN activity and reward rate is highly correlated in both regions of the striatum in both rewarded and unrewarded trials to suggest that iSPN activity is uncorrelated with dopamine.

The rationale that the "analysis was designed to evaluate the overall response pattern within this period rather than moment-to-moment changes" is perhaps their way of "not seeing the trees for the forest".

The manuscript is much improved, and there are still some major concerns particularly concerning the use of "reward rate" as used in this study. Although the authors have put in words to somewhat address prior concerns, the bottom line conclusion most readers will likely take is that DA increases iSPN activity.

Review of

Reward expectation enhances action-related activity of nigral dopaminergic and two striatal output pathways

A: Thank you for your feedback on the revised version of our manuscript.

We would like to address your concern regarding the interpretation of our findings related to dopamine and iSPN activity. It is evident that the revised manuscript has led to a clearer distinction between the roles of dopamine and other potential inputs in modulating iSPN activity. We acknowledge your point that the previous version might have suggested a stronger role for dopamine than intended. Your feedback has prompted us to refine our explanations and ensure that the conclusions drawn are aligned with the actual data.

Upon reevaluation of our study's results, we fully recognize that dopamine might not be the primary driver of iSPN activity increasing following higher reward rate. We agree with your observation that the data in Figure 5c indicate a potential dissociation between iSPN activity and dopamine release timing. As pointed out, our original interpretation did not adequately reflect this finding. We have revised the text to more accurately convey that iSPN activity may not be primarily driven by dopamine and could be influenced by other inputs, such as cortical and thalamic inputs. We addressed this issue in the Abstract (L-56) and Discussion (L-398, 403, 416).

Our explanation that the analysis was designed to evaluate the overall response pattern within a longer time period may be considered as a simplification, and we appreciate your suggestion to clarify this point further. We have taken steps to elaborate on this aspect, addressing the potential limitation and its implications for the interpretation of our results (L-471, 477, 485).

The reviewer's comments are shown below:

The manuscript is much improved. The experimental design is considerably easier to understand, and the description of various measures much clearer. A major criticism was that in the prior versions, the authors suggested that dopamine increased activity of D2 expressing iSPNs without evidence. The authors are now stating that "We do not consider that iSPN activity is dependent on DA", but rather 'on recent rewards', and iSPN may be modulated by other inputs; i.e. cortex thalamus, etc.". However, as written are likely to draw the conclusion that dopamine increases iSPN activity mainly because that is the way the study is structured.

The study first finds that "dopamine" activity and release is correlated with "reward rate" and then that "reward rate" is correlated with iSPN activity.

- b. The increase in iSPN activity tied to the initiation of movement (in fig 5c) precedes the increase in DA release in the striatum (Fig 2c middle) so that it is hard to make the case that iSPN activity is "dependent" on DA.**

We do not consider that 'iSPN activity is dependent on DA', but rather 'on recent rewards', and iSPN may be modulated by other inputs; i.e. cortex thalamus, etc. (please see the above responses, too). We apologize if our statements were not clear enough in the manuscript.

We agree with the reviewer's important point that iSPN activity preceded DA release. This is exactly the evidence that iSPN activity is not driven solely by DA. We come to consider that inputs from other regions, such as the cortex or thalamus, might play significant roles in modulating the activities of both striatum and dopaminergic neurons in the midbrain during this time frame (see Fig. 6).

We have made this point clear in the Results (L.309-312).

309 nigral and striatal neurons except for DMS–SNc neurons (Fig. S7). Importantly, the activation of those neurons preceded dopamine release in 310 the striatum (e.g., Fig.

311 2c vs. Figs. 3c, 4c, and 5c), suggesting possible common inputs other than 312 dopamine.

The data in Figure 5c shows that iSPN activity increases prior to DA release, which is what is stated, but then it is unclear what is meant by “suggesting possible common inputs other than dopamine”. It actually suggests that iSPN activity might not be driven by dopamine at all. The text reads as though dopamine is the primary player in the patterns of activity, while the data suggest it might be a secondary player.

- c. For the “outcome tone” period the highest level of activity is in the 100 msec following tone delivery after which the activity in iSPNs decreases (Fig 5c). In this time period striatal DA release does not start to increase for 150 msec or so into the time period, at which time the activity of iSPNs is suppressed. This does not support the idea that DA enhances iSPN activity.

The reviewer is correct in stating that the onset of iSPN activity reduction appears to coincide with the time point when striatal DA release begins to increase following the outcome tone, Results (L.306). The temporal dynamics indeed presents a nuanced scenario, which we did not explicitly address in our initial manuscript.

However, it is important to clarify that our claim is not that dopamine directly causes an instantaneous decrease in iSPN activity. Rather, we observed a positive correlation between dopamine release and iSPN activity in relation to the reward information over a broader time window (recent 5 trials), not on a moment-to-moment basis; i.e. only last or current trial.

Additionally, it is worth mentioning that dopamine effects are multifaceted and context-dependent. Dopamine can have both fast and slow effects, impacting different aspects of neuronal function, including excitability, synaptic plasticity, and long-term changes in gene expression.

- d. The “reward rates” correlations to activity levels are calculated for the 500 msec time from the beginning of the “outcome tone”, during this time the activity levels go through various phases, first increasing briefly and then decreasing. An average number obtained would not necessarily reflect the effect of DA.**

We understand this point about the changing nature of neuronal activity during the 500 msec window following the onset of the "outcome tone".

However, our analysis was designed to evaluate the overall response pattern within this period rather than moment-to-moment changes. We acknowledge that the instantaneous activity levels do vary during this period, and thus, the averaged activity may not precisely reflect the effect of dopamine at every single point within this time window. However, it serves as an approximate measure to compare the overall response magnitude across different trials and conditions, allowing us to observe general trends and correlations with reward rate.

The most confusing part of this study is the data shown in Figure 5. As stated iSPN activity in rewarded trials increases prior to DA release, and then decreases. This suggests and is stated that after reward outcomes iSPN activity is suppressed. However, then in 5d and e, measures are shown that there is a correlation of iSPN activity with reward rate. The authors explain that while there the relationship between iSPN activity and the time course of the behavior “presents a nuanced scenario” that is due to the measure being averaged over a longer time period (that encompasses both the very transient increase in iSPN activity that appears unrelated to dopamine and the longer period of suppression that occurs after dopamine release begins). One might interpret the finding that the iSPN activity and reward rate is highly correlated in both regions of the striatum in both rewarded and unrewarded trials to suggest that iSPN activity is uncorrelated with dopamine.

The rationale that the “analysis was designed to evaluate the overall response pattern within this period rather than moment-to-moment changes” is perhaps their way of “not seeing the trees for the forest”.

The manuscript is much improved, and there are still some major concerns particularly concerning the use of “reward rate” as used in this study.

We understand that our manuscript can be improved by providing a more nuanced discussion on these aspects, and we have revised the Discussion (L.353-364) to more clearly communicate these points.

d. The “reward rates” correlations to activity levels are calculated for the 500 msec time from the beginning of the “outcome tone”, during this time the activity levels go through various phases, first increasing briefly and then decreasing. An average number obtained would not necessarily reflect the effect of DA.

We understand this point about the changing nature of neuronal activity during the 500 msec window following the onset of the "outcome tone".

However, our analysis was designed to evaluate the overall response pattern within this period rather than moment-to-moment changes. We acknowledge that the instantaneous activity levels do vary during this period, and thus, the averaged activity may not precisely reflect the effect of dopamine at every single point within this time window. However, it serves as an approximate measure to compare the overall response magnitude across different trials and conditions, allowing us to observe general trends and correlations with reward rate.

e. For the “go cue”, “movement initiation” and “outcome tone” period action related activity is reported to be positively modulated by dopamine in both the “rewarded” and “nonrewarded” trials, which do not appear to be directly correlated with DA release in the striatum. This does not support the idea that such activity is dependent on DA.

We would like to clarify that our claim is not that iSPN activity is directly dependent on DA release. We agree with the reviewer’s pointing out that the correlations we see between dopamine levels and iSPN activity do not necessarily mean that one directly causes the other. The relationship we found might be indicative of an indirect or modulatory role of dopamine, or they might both be influenced by a common third factor.

In fact, as the reviewer has mentioned, other inputs, such as from the cortex or thalamus, may also be important for modulating the activity of iSPNs during the periods (see Fig. 6). We should have made this point clearer in our initial manuscript, and we have revised it accordingly to more precisely describe the nature of the relationship we observed between dopamine levels and iSPN activity, Results (L.366-382).